# Numerical modeling of stresses and deformation in the Zagros-Iranian plateau region

Srishti Singh [1] and Radheshyam Yadav [1]

[1]CSIR-National Geophysical Research Institute, Hyderabad, India-500007

**Correspondence:** Radheshyam Yadav(shamgpbhu@gmail.com)

**Abstract.** Zagros orogeny System resulted due to collision of the Arabian plate with the Eurasian plate. The region has the ocean-continent subduction and continent-continent collision; and convergence velocity shows variation from east to west. Therefore, this region shows the complex tectonic stress and a wide range of diffuse or localized deformation between both plates. The in-situ stress and GPS data are very limited and sparsely distributed in this region, therefore, we performed a numerical simulation of the stresses causing deformation in the Zagros-Iran region. The deviatoric stresses resulting from the variations in lithospheric density and thickness; and those from shear tractions at the base of the lithosphere due to mantle convection were computed using thin-sheet approximation. Stresses associated with both sources can explain various surface observations of strain rates, $S_{Hmax}$, and plate velocities; thus, suggesting a good coupling between lithosphere and mantle in most parts of Zagros and Iran. As the magnitude of stresses due to shear tractions from density-driven mantle convection is higher than those from lithospheric density and topography variations in the Zagros-Iranian plateau region, mantle convection appears to be the dominant driver of deformation in this area. However, the deformation in the east of Iran is caused primarily by lithospheric stresses. The plate velocity of the Arabian plate is found to vary along the Zagros belt from the north-northeast in the southeast of Zagros to the northwest in northwestern Zagros, similar to observed GPS velocity vectors. The output of this study can be used in seismic hazards estimations.

## 1 Introduction

Zagros mountains are a part of the Alpine-Himalayan belts that originated due to the Arabian plate colliding with southern boundary of the Eurasian plate. This collision resulted from the closing of the Neotethys Ocean, and formed Zagros fold and thrust belt (Agard et al., 2005, 2011; Alavi, 1980; Mouthereau et al., 2012). The Zagros mountains extend from the eastern part of Anatolia for over 1500 km in the NW-SE direction till the Makran subduction zone, showing large-scale diffuse deformation. Despite the first-order characteristics of the active deformation and present-day kinematics of Zagros orogen being relatively well understood (Allen et al., 2011; Le Dortz et al., 2009; Reilinger et al., 2010; Vernant et al., 2004; Walker, 2006), the timing of the collision is debated. The timing of collision ranges from Cretaceous (Alavi, 1994; Mohajjel and Fergusson, 2000) to Miocene (Berberian and King, 1981) or Eocene (Allen and Armstrong, 2008; Jolivet and Faccenna, 2000). However, there has been an increasing consensus on Late Eocene to Oligocene for the onset of collision (Jolivet and Faccenna, 2000; Agard et al., 2005, 2011; Vincent et al., 2005; Ballato et al., 2011; Mouthereau et al., 2012; Koshnaw et al., 2019). The Arabia-Eurasia

collision zone is a tectonically active region, where ongoing convergence is accommodated by distributed shortening across the Zagros Mountains and the northern and eastern margins of the Iranian Plateau and the southern Caspian Sea. The rate of convergence of Arabia relative to Eurasia also varies significantly, decreasing from 36 mm/yr in the east to 16 mm/yr in the west (Figure 1). The diverse structures, tectonic history, and convergence velocity variations in the Zagros-Iran plateau region lead to variable tectonic stresses and deformations, thus making it the focus of various geophysical, geological, and geodesy studies (Engdahl et al., 2006; Hatzfeld et al., 2010; Khorrami et al., 2019; Masson et al., 2006; Tunini et al., 2016, 2017). Based on earthquake focal mechanisms, fault slip, and GPS velocities, the Zagros-Iran region has been categorized as a highly seismic region; thus a better constraint on stresses and deformation in this region may be helpful in disaster mitigation studies.

Generally, tectonic stress refers to the forces acting on the Earth's crust that cause it to deform or undergo changes and it's classified by the first, second and third order on the spatial scale (Heidbach et al., 2007; Zoback, 1992). The first-order stresses originate due to the plate boundaries force like ridge push, slab pull and continental collisional; and second-order stresses by the rifting, isostasy and deglaciation. Moreover, third-order stresses are caused by local sources like interaction faults systems, topography and density heterogeneity. Therefore, to understand the origin of these stresses, in-situ stress measurements are done using the focal mechanism inversion, wellbore breakouts, hydraulic fracturing and overcoring, and compiled under the word stress map project. However, in-situ stress data are sparsely distributed and limited, so numerical modeling plays an important role in understanding the kinematics and dynamics of the Zagros-Iran region. Numerical modeling of tectonic stresses and deformation is generally conducted in two approaches (1) using 2D and 3D geometrical structure, plate boundary forces like ridge push, slab pull and continents collision forces and rheological properties like Young's modulus, Poisson's ratio, viscosity, density etc. (Coblentz and Sandiford, 1994; Dyksterhuis and Müller, 2008; Koptev and Ershov, 2010; Richardson et al., 1976; Yadav and Tiwari, 2018), and, (2) considering Gravitational Potential energy and shear tractions from mantle convection with thin sheet approximation (Bird, 1998; Flesch et al., 2001; Ghosh and Holt, 2012; Lithgow-Bertelloni and Guynn, 2004; Singh and Ghosh, 2020).

There are various studies that have tried to investigate present-day stresses and deformations of the Zagros-Iranian plateau region using focal mechanism inversions, GPS data and numerical modeling. The stresses were computed through the inversion of focal mechanisms in areas like the Zagros fold-and-thrust belt (Nouri et al., 2023; Sarkarinejad et al., 2018; Yaghoubi et al., 2021), Zagros-Makran transition zone (Ghorbani Rostam et al., 2018), western Zagros (Navabpour et al., 2008), NW Iran-SE Turkey (Mokhoori et al., 2021), NE Lut Block, Eastern Iran (Rashidi et al., 2022; Raeesi et al., 2017), and the South Caspian (Jackson et al., 2002). The GPS studies also provided constraints on the present-day deformation in Zagros-Makran transition zone (Bayer et al., 2006), Makran subduction zone (Frohling and Szeliga, 2016), Iran (Khorrami et al., 2019; Masson et al., 2006, 2007; Vernant et al., 2004; Walpersdorf et al., 2014), and Nubia–Arabia–Eurasia plate system (Reilinger and McClusky, 2011). Sobouti and Arkani-Hamed (1996) studied the large scale tectonic processes of the region and reproduced observed faulting patterns by considering highly rigid central Iran and the South Caspian Sea using a viscous thin-sheet approximation. On the other hand, Md and Ryuichi (2010) used finite element modeling (FEM) to analyze the neotectonic stress field of Zagros and adjoining area and showed N-S/NNE-SSW oriented $S_{Hmax}$ in Lurestan and eastern Zagros Simple Folded Belt, NW-SE around Main Recent fault (MRF) and in northern High Zagros Faults (HZF). Further, the kinematic model by Khodaverdian

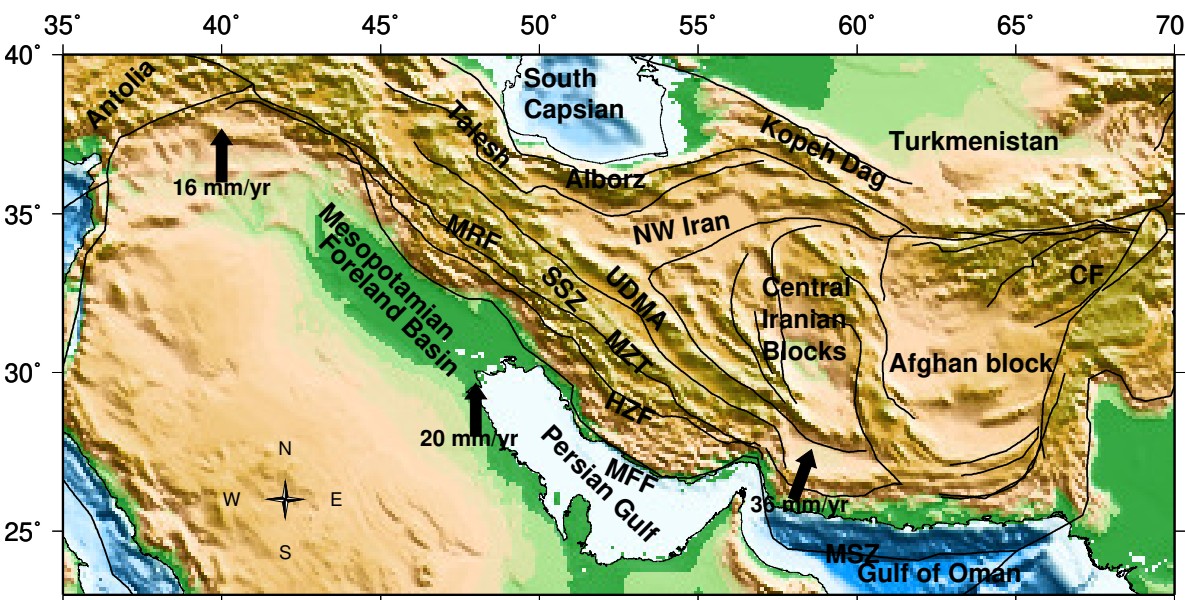

**Figure 1.** Tectonic overview of Central Eurasia. Abbreviations: CF: Chaman Fault; MSZ: Makran Subduction Zone; MZT: Main Zagros Thrust; HZF: High Zagros Fault; MFF: Mountain Front Fault; SSZ: Sanandaj Sirjan Zone; UDMA: Urumieh-Dokhtar Arc; MRF: Main Recent Fault.

et al. (2015) provided constraints on fault slip rates, plate velocities and seismicity of the Iranian Plateau. Most of the deformation studies done in this region focus on different tectonic fragments of the Arabia-Eurasia collision zone. Moreover, the previous studies do not include the role of shear tractions associated with mantle convection in affecting the deformation and stresses in the Zagros-Iran regions.

In this study, we investigate the stress and deformation in the entire Zagros-Iranian Plateau region to constrain the forces acting in this region with gravitational potential energy (GPE) and shear traction of mantle tractions. We will use a thin viscous sheet model based on Flesch et al. (2001) to compute various deformation parameters such as deviatoric stresses, strain rates, most compressive principal stress ($S_{Hmax}$), and plate velocities within the Zagros-Iran region.

## 2   Tectonic and Geology

The evolution of the Zagros mountain belt is a direct consequence of the continental collision between the Arabian and Eurasian plates. Zagros is located at the northeastern margin of the Arabian plate, trending in the southwest direction (Figure 1). It is bounded by the Main Zagros thrust (MZT) in the northeast, while it joins the Tauras mountains in southern Turkey in the northwest. In the southeast, N-S trending Minab-Zandan fault zone separates Zagros from the Makran range. Outer Zagros are the young folded mountains in the southwest of the orogeny (Falcon, 1974; Sattarzadeh et al., 2002). High Zagros fault (HZF)

separates highly deformed metamorphic rocks of inner Zagros from Simply folded mountains of outer Zagros (Hatzfeld and Molnar, 2010; Hatzfeld et al., 2010). Inner Zagros are bounded by MZT in northeast and are dominated by thrust faulting,

possibly due to compression during the Late Cretaceous (Alavi, 1980). The northwestern Zagros is separated from central Zagros by a north-south trending strike-slip zone of deformation, known as Kazerun Fault System (KFS) (Authemayou et al., 2005).

Zagros mountains were formed between ∼35 and ∼23 Ma due to the convergence of the Arabian platform beneath the central Iranian crust (Agard et al., 2005; Ballato et al., 2011; Mouthereau et al., 2012). The Arabian plate moves towards Eurasia with a plate velocity of 22-35 mm/yr (DeMets et al., 1990; McClusky et al., 2000; Jackson et al., 2002; McQuarrie et al., 2003; Reilinger et al., 2006) in N-S to NNE direction. The convergence of two rigid plates of Arabia and Eurasia leads to a zone of widespread deformation in the form of the high plateaus of Iran. Iranian plateau extends from the Caspian Sea and the Kopeh

Dagh range in the north to the Zagros Mountains in the west. Iranian plateau is bounded by the Persian Gulf and Hormuz Strait in the south and the country's political borders on the eastern side. Several tectonic processes such as intracontinental collisions, subduction along the Makran and the transition from Zagros fold-thrust belt to the Makran subduction zone contribute to the complex tectonics of the Iranian plateau.

During the last few decades, various geophysical studies (receiver functions, deep seismic, GPS and tomographic) have been

carried out in the Zagros-Iran region to investigate the structure and deformation in this region. The southeastern Zagros accommodate the convergence between Arabia and Eurasia by pure shortening occurring through high-angle ($30° - 60°$) reverse faults that are perpendicular to the belt (Hessami et al., 2006; Irandoust et al., 2022; Walpersdorf et al., 2006). On the other hand, oblique convergence in central and northern Zagros is partitioned into a strike-slip component that is accommodated on MRF and shortening occurring across the belt (Jackson et al., 2002; Talebian and Jackson, 2002). Zagros is separated from

Makran subduction zone (MSZ) by Minab-Zendan-Palami (MZP) fault ($54° - 58°E$), which is a right-lateral strike-slip fault (Bayer et al., 2006). East of MZP shows significant shortening that is accommodated through the subduction in MSZ. Due to the difference between convergence rates, a shearing occurs in eastern Iran, which is accommodated by the N-S trending faults bounding the Lut block. In northern Iran, fold and thrust belt of Alborz accommodates a quarter of the Arabia-Eurasia convergence Irandoust et al. (2022). The oblique convergence in eastern Alborz is also partitioned into shortening at the south-

ern boundary and a left-lateral component across the mountain belt (Irandoust et al., 2022; Khorrami et al., 2019; Tatar and Hatzfeld, 2009). Alborz mountains extend into Talesh in the west, which shows thrust faulting on nearly flat faults. Kopeh-Dagh range in the northeast accommodates the Arabia-Eurasia convergence through N-S shortening on major thrust faults in south.

## 3 Modeling

 ## 3.1 Equations

To model the present-day stresses causing deformation in the Zagros-Iranian plateau due to the Arabia-Eurasia collision, we solve the force balance equation, considering the thin sheet approximation.

$$\frac{\partial \sigma_{ij}}{\partial x_j} + \rho g_i = 0 \tag{1}$$

Here $\sigma_{ij}$, $x_j$, $\rho$, and $g_i$ indicate the $ij^{th}$ component of the total stress tensor, $j^{th}$ coordinate axis, density and acceleration due to gravity respectively (England and Molnar, 1997; Ghosh et al., 2013b).

In the above equation, total stress, $\sigma_{ij}$ is substituted by deviatoric stress using the following relation:

$$\tau_{ij} = \sigma_{ij} - \frac{1}{3} \sigma_{kk} \delta_{ij} \tag{2}$$

In the above equation, the Kronecker delta and mean stress are denoted by $\delta_{ij}$ and $\frac{1}{3}\sigma_{kk}$ respectively. The force balance equation (1) is integrated up to the base of lithospheric sheet (L), resulting in following full horizontal force balance equations:

$$\frac{\partial \overline{\tau}_{xx}}{\partial x} - \frac{\partial \overline{\tau}_{zz}}{\partial x} + \frac{\partial \overline{\tau}_{xy}}{\partial y} = -\frac{\partial \overline{\sigma}_{zz}}{\partial x} + \tau_{xz}(L) \tag{3}$$

$$\frac{\partial \overline{\tau}_{yx}}{\partial x} + \frac{\partial \overline{\tau}_{yy}}{\partial y} - \frac{\partial \overline{\tau}_{zz}}{\partial y} = -\frac{\partial \overline{\sigma}_{zz}}{\partial y} + \tau_{yz}(L) \tag{4}$$

In equation (3) and (4), the over bars indicate integration over depth. Both equations (3 and 4) contain the first term representing horizontal gradients of GPE per unit area on the right hand side. On the other hand, the shear tractions at the lithosphere base
(L) arising due to mantle convection are denoted by $\tau_{xz}(L)$ and $\tau_{yz}(L)$ (Ghosh et al., 2009).

Both of the force balance equations (3 & 4) were solved using the finite element technique (Flesch et al., 2001; Ghosh et al., 2009, 2013b; Singh and Ghosh, 2019, 2020) for a 100 km thick lithosphere of varying strength (Figure S1a). The laterally varying viscosities for the lithosphere were assigned from Singh and Ghosh (2020). After solving these equations, we obtained the horizontal deviatoric stresses, $S_{Hmax}$, strain rates as well as plate velocities and compared them with observations.

The quantitative comparison between predicted and observed $S_{Hmax}$ axes (Figure 3a) was performed by computing the misfit given by $sin\theta(1+R)$ (Ghosh et al., 2013a; Singh and Ghosh, 2019, 2020), where R represents the quantitative difference between stress regimes of observed and predicted $S_{Hmax}$, while $\theta$ denotes the angular difference between both. Hence, this misfit accounts for both the angular and regime misfits with values lying between 0 and 3.

The correlation between predicted deviatoric stresses and GSRM strain rates (Figure 3b) (Flesch et al., 2007; Ghosh et al., 2013b; Singh and Ghosh, 2019, 2020) is given by the following equation:

$$-1 \leq \sum_{areas} (\varepsilon.\tau)\Delta S / \left( \sqrt{\sum_{areas} (E^2)\Delta S} * \sqrt{\sum_{areas} (T^2)\Delta S} \right) \leq 1 \tag{5}$$

where $E = \sqrt{\dot{\varepsilon}_{\phi\phi}^2 + \dot{\varepsilon}_{\theta\theta}^2 + \dot{\varepsilon}_{rr}^2 + \dot{\varepsilon}_{\phi\theta}^2 + \dot{\varepsilon}_{\theta\phi}^2} = \sqrt{2\dot{\varepsilon}_{\phi\phi}^2 + 2\dot{\varepsilon}_{\phi\phi}\dot{\varepsilon}_{\theta\theta} + 2\dot{\varepsilon}_{\theta\theta}^2 + 2\dot{\varepsilon}_{\phi\theta}^2}$, $T = \sqrt{\tau_{\phi\phi}^2 + \tau_{\theta\theta}^2 + \tau_{rr}^2 + \tau_{\phi\theta}^2 + \tau_{\theta\phi}^2}$ $= \sqrt{2\tau_{\phi\phi}^2 + 2\tau_{\phi\phi}\tau_{\theta\theta} + 2\tau_{\theta\theta}^2 + 2\tau_{\phi\theta}^2}$, and $\varepsilon.\tau = 2\dot{\varepsilon}_{\phi\phi}\tau_{\phi\phi} + \dot{\varepsilon}_{\phi\phi}\tau_{\theta\theta} + \dot{\varepsilon}_{\theta\theta}\tau_{\phi\phi} + 2\dot{\varepsilon}_{\theta\theta}\tau_{\theta\theta} + 2\dot{\varepsilon}_{\phi\theta}\tau_{\phi\phi}$. In the above equation, the second invariants of the strain rate and stress tensors are denoted by $E$ and $T$. GSRM strain rates, area and predicted deviatoric stresses are represented by $\dot{\varepsilon}_{ij}$, $\Delta S$, and $\tau_{ij}$ respectively. We also get the relative plate velocities and strain rates as output from models. However, to calculate the absolute plate velocities and strain rates, we require absolute viscosity values. We compute the scaling factor for relative viscosities by placing the predicted velocities in a no-net-rotation (NNR) frame, such that $\int (v \times r)dS = 0$ and minimizing the misfit between the predicted dynamic velocities and those from Kreemer et al. (2014). Here $v$ denotes the horizontal surface velocity at position $r$ and $S$ is the area over the Earth's surface (see Ghosh et al. (2013b) for details).

## 3.2 Crustal Models

In the right hand side of equations (3 & 4), the first term represents the vertically integrated vertical stress. It is computed and integrated from the top of variable topography up to depth L (100 km) (England and Molnar, 1997; Flesch et al., 2001; Ghosh et al., 2013b; Singh and Ghosh, 2019, 2020) using the following relation:

$$\overline{\sigma}_{zz} = -\int_{-h}^{L} \left[ \int_{-h}^{z} \rho(z')gdz' \right] dz = -\int_{-h}^{L} (L-z)\rho(z)gdz \tag{6}$$

where $\rho(z)$, $L$ and $h$ denote density, the depth to the lithosphere base (100 km) and topographic elevation respectively. $z$ & $z'$ are variables of integration and $g$ represents the acceleration due to gravity. We also calculated the stresses for thicker lithosphere (L=150 km and L=200 km) as studies have shown a much thicker lithosphere in the region (Robert et al., 2017; Tunini et al., 2017) (Figure S2a-f).

The right hand side of equation 6 is given by the negative of GPE per unit area. To calculate GPE and the stresses associated with it, we used three global crustal models, CRUST2.0 (Bassin et al., 2000), CRUST1.0 (Laske et al., 2013), and LITHO1.0 (Pasyanos et al., 2014). The upper crust thickness lies within 15-20 km in the Zagros-Iran region for CRUST2 model (Figure 2a). However, the thickness of the upper crust in the Zagros-Iranian region is much higher for CRUST1 and LITHO1 (> 25 km) (Figures 2b & c). The Zagros-Iran region has a thicker middle crust (> 20 km) in the case of both CRUST2 and LITHO1 models (Figures 2d & f), while CRUST1 shows a much thinner middle crust (< 12 km) in this region (Figure 2e). The lower crust in the Zagros-Iran region is found to be very thin (< 10 km) for all three models (Figure 2g-i).

The density variations in the study area are minimal for CRUST2 model. CRUST2 also shows the highest average density in all three layers (>2.7 $g/cm^3$) (Figure 2j,m,p). CRUST1 also indicates an average density of ~2.72 $g/cm^3$ in the Zagros-Iran

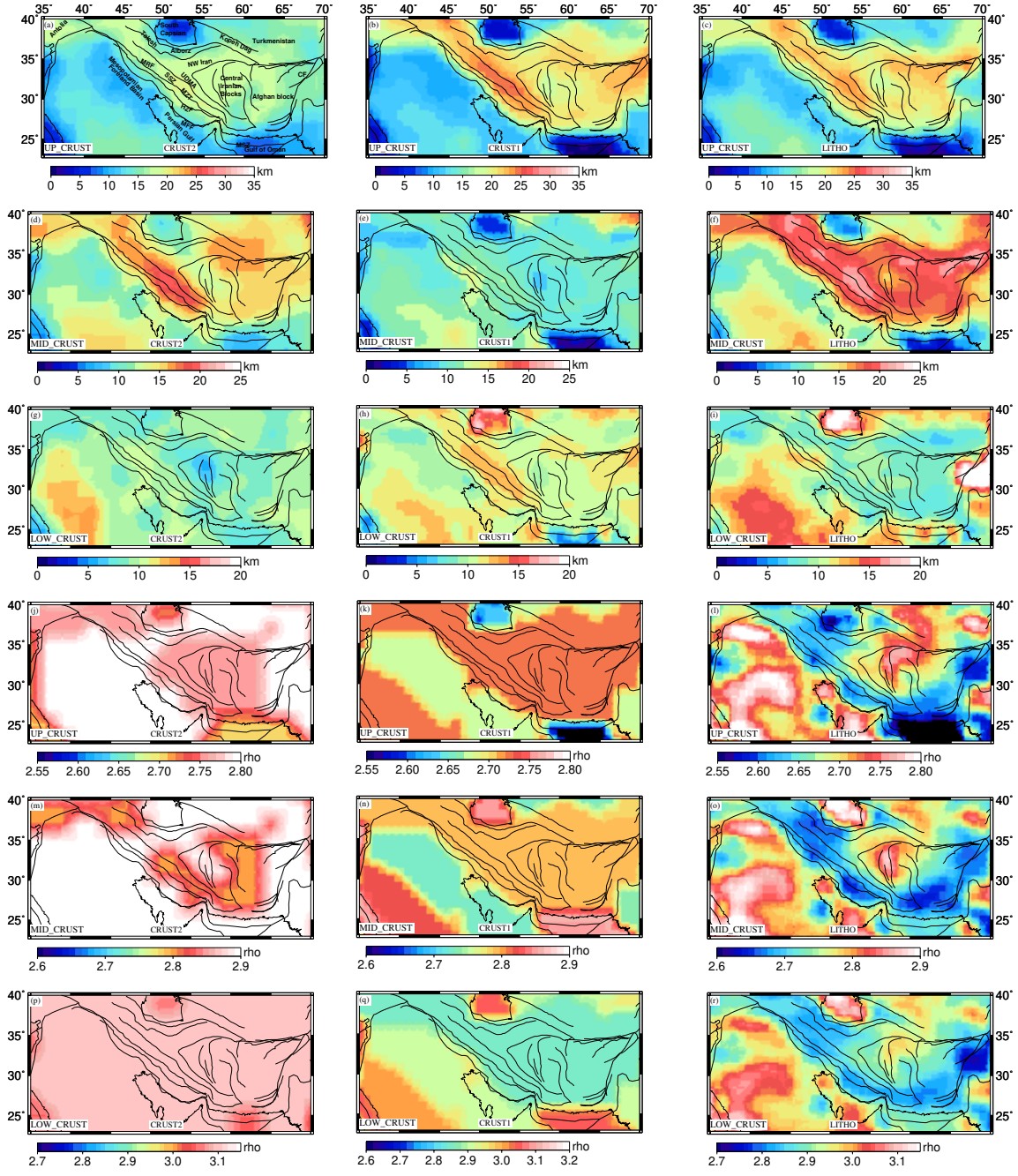

**Figure 2.** Thickness and density variations of different layers in all three crustal and lithospheric models: CRUST2(Left panel), CRUST1(Middle Panel) and LITHO1(Right panel).

region for the upper crust (Figure 2k). The middle and lower crustal layers of CRUST1 show average densities of 2.80 $g/cm^3$ and $\sim$2.85 $g/cm^3$, respectively (Figure 2n,q). LITHO1 model shows the lowest average density in the study area for all three layers (Figure 2l,o,r). The upper crust of LITHO models shows an average density of $\sim$2.65 $g/cm^3$. Central Iran block has a relatively denser upper crust ($\sim$ 2.75 $g/cm^3$), while the density decreases to $\sim$ 2.62-2.64 $g/cm^3$ near the Zagros region. Similar patterns of density variations are observed in the middle and lower crust of LITHO1 model ((Figure 2o,r). Such differences in thickness and density data lead to varying GPE values, and hence subsequently, different stresses.

### 3.3 Mantle Convection

We ran mantle convection models using HC (Hager and O'Connell, 1981). HC is a semi-analytical mantle convection code that uses density anomalies derived from seismic tomography models and radial viscosity as inputs. Here, we considered four global mantle convection models, S40RTS (Ritsema et al., 2011), SAW642AN (Mégnin and Romanowicz, 2000), 3D2018_S40RTS and S2.9_S362 to infer the mantle density anomalies. 3D2018_S40RTS is a merged model of SV wave upper mantle tomography model, 3D2018_Sv given by Debayle et al. (2016), and S40RTS. S2.9 is a global tomography model of the upper mantle with higher resolution, which is given by Kustowski et al. (2008b). We merged this model with the global shear wave velocity model, S362ANI (Kustowski et al., 2008a) to obtain the merged tomography model of S2.9_S362. We used two different radial viscosity structures, namely GHW13, which is the best viscosity model from Ghosh et al. (2013b), and SH08 given by Steinberger and Holme (2008). GHW13 is a four layered viscosity structure, with a highly viscous lithosphere ($\sim 10^{23}$ Pa-s). The viscosity drops to $\sim 10^{20}$ Pa-s in the asthenosphere, which again increases to $\sim 10^{21}$ Pa-s in the upper mantle and $\sim 10^{22}$ Pa-s in the lower mantle (Figure S1b). On the other hand, the viscosity in SH08 model increases gradually with depth and it has a slightly weaker lithosphere as compared to GHW13. It has the highest viscosity value of $10^{23}$ Pa-s around 2000-2300 km depth, and significantly lower viscosity for D" layer (Figure S2b). GHW13 viscosity model performed slightly better than SH08 in fitting the observed parameter, thus we have shown results from the same throughout this paper. However, we have also included the predicted results and their fit to the observables in the supplementary section (Table S1).

### 3.4 Data

To have better constraints on this study's models, we also estimated $S_{Hmax}$ (most compressive horizontal principal axes) orientations as well as plate velocities. Various deformation indicators such as $S_{Hmax}$ orientations from the World Stress Map (WSM) (Heidbach et al., 2016), strain rates and plate velocities from Global Strain Rate Model (Kreemer et al., 2014) were used to perform a quantitative comparison with the predicted results of this study (Figure 3).

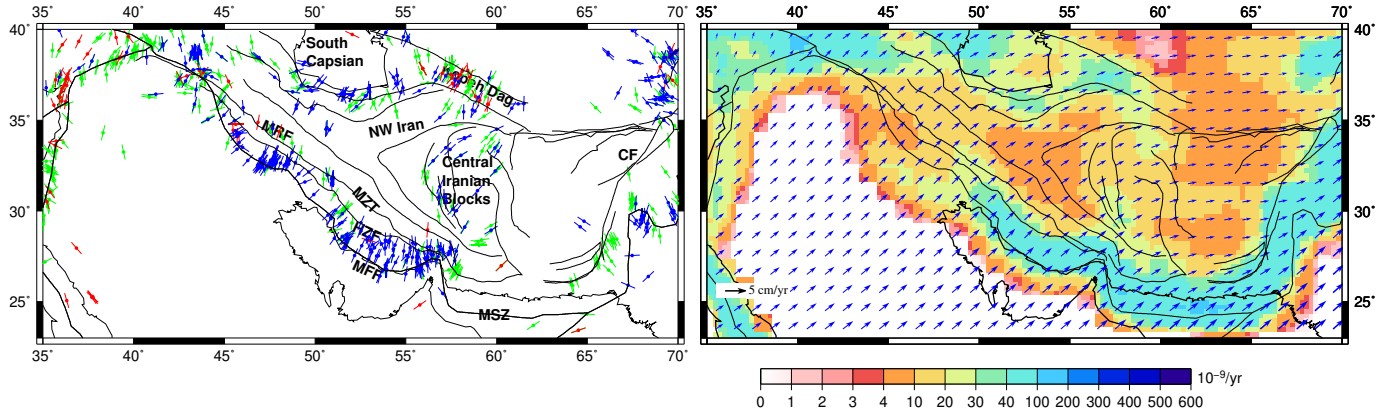

**Figure 3.** (a) Most compressive horizontal principal axes ($S_{Hmax}$) from WSM (Heidbach et al., 2016). Red indicates normal fault regime, and blue indicates thrust regime, whereas green denotes strike-slip regime, (b) Observed plate velocities in a no-net-rotation frame of reference from Kreemer et al. (2014) plotted on top of the second invariant of strain rate tensors obtained from Kreemer et al. (2014) plotted on $1° \times 1°$ grid.

WSM data is a global database of the crustal stress field obtained from various sources such as focal mechanisms; geophysical logs of borehole breakouts and drilled induced fractures; engineering methods such as hydraulic fractures and overcoring; and geological indicators that are obtained from fault slip analysis and volcanic alignments. These data have been assigned quality ranks from A to E based on the accuracy range. A-type data suggests that the standard deviations of $S_{Hmax}$ orientations are within $\pm 15°$ range, $\pm 20°$ for B-type, $\pm 25°$ for C-type and $\pm 40°$ for D-type. However, E-type indicates the data records are either incomplete or from non-reliable sources or the accuracy is $> \pm 40°$. This study uses A-C quality stress data records (Figure 3a). Observed $S_{Hmax}$ axes are aligned in NNE-SSW directions in Zagros with dominant thrust faulting. NW and Central Iran show some strike-slip mode of deformation with NE-SW compressional directions.

The strain rates and plate velocities are taken from GSRM v2.1 model (Kreemer et al., 2014) (Figure 3b). GSRM v2.1 provides a global data set of strain rates and plate motions that are determined using $\sim 22,500$ geodetic plate velocities. Higher strain rates are observed along the simply folded mountains ($\sim 40 - 100 \times 10^{-9}/yr$). Most of Iran shows strain rates between $4 - 10 \times 10^{-9}/yr$. The plate motions used in our study for comparing with predicted velocities are given in a no-net-rotation (NNR) frame interpolated on a $1° \times 1°$ grid. The velocity vectors show an eastward motion in the study area, which becomes nearly E-W in Afghan Block (Figure 3b).

## 4    Results

### 4.1    Stress and deformation due to GPE

Three crustal models (CRUST1.0, CRUST2.0 and LITHO1.0) were used to compute GPE within the study region. The second invariant of stress computed using GPE lies within $\sim$10-12 MPa along the Zagros for CRUST2 and CRUST1 models (Figure 4a,c). LITHO1 model predicts larger stress magnitudes along Zagros (Figure 4e). NE-SW compressional stresses are observed

along the frontal faults of Zagros (MFF) (Figure 1a,c). The central part of Zagros thrust faults (MZT) shows the strike-slip mode of faulting for nearly all three models (Figures 4 & 4b,d & f). The strike-slip regime further extends into Sanandaj-Sirjan Zone (SSZ) while lies north of MZT for CRUST2 and LITHO1 model (Figure 4b,f), while it transitions to thrust type of deformation in the north of MZT for CRUST1 (Figure 4d). The Urmia-Dokhtar Magmatic Arc (UDMA) and central Iran also show the strike-slip mode of faulting for CRUST2 and LITHO1. The north of MRF shows tension for CRUST2 model, while CRUST1 predicts this area to be predominantly strike-slip. On the other hand, the entire region shows significant compression for LITHO1 model.

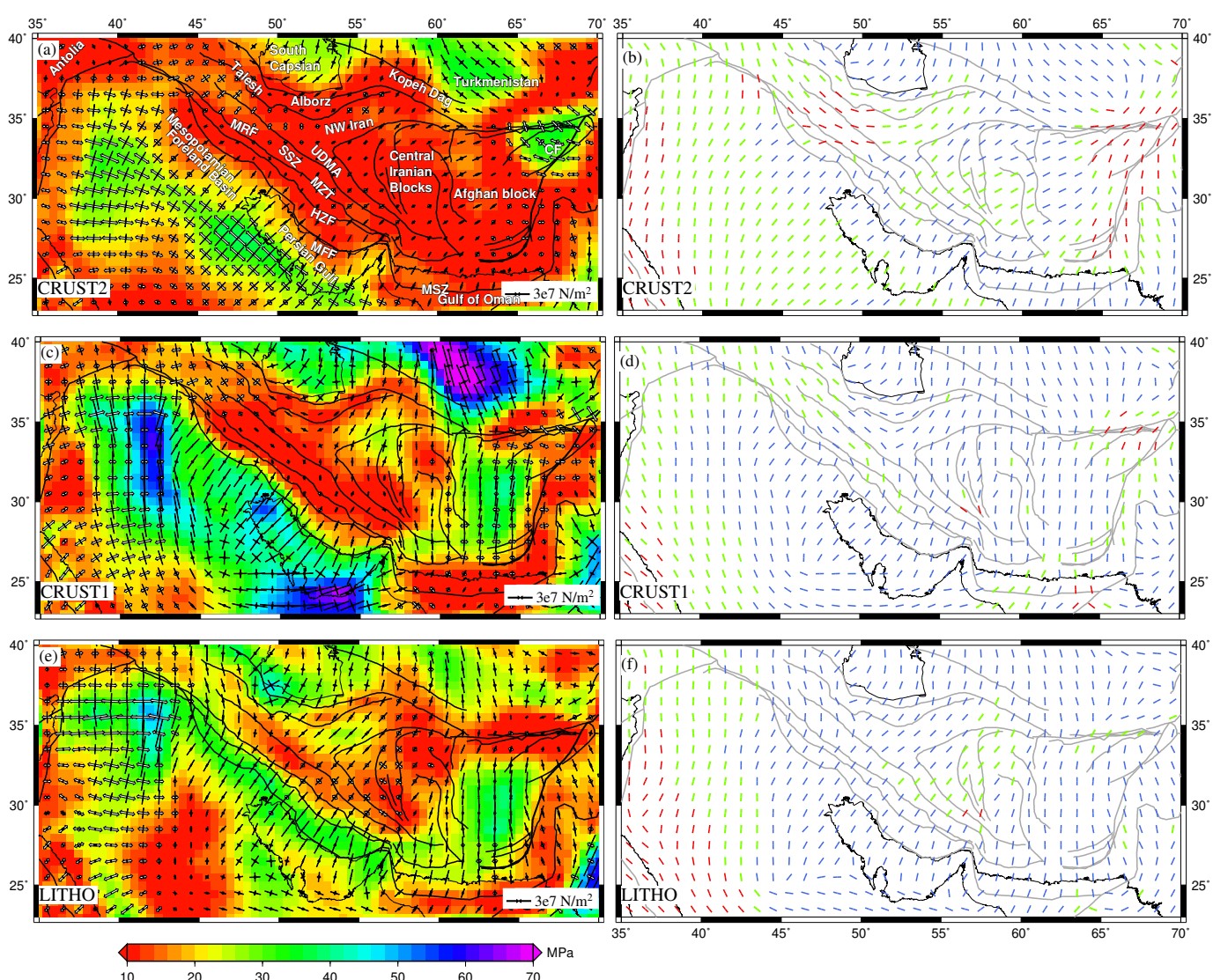

**Figure 4.** (Left Panel) Deviatoric stresses predicted from GPE variations, plotted on top of their second invariants. The compressional stresses are denoted by solid black arrows, while white arrows show tensional stresses. $S_{Hmax}$ axes predicted from GPE variations are plotted in the right panel. Red denotes the tensional regime, blue is for thrust and green is for the strike-slip regime.

We compared predicted $S_{Hmax}$ from our three GPE only models to observed $S_{Hmax}$ orientations and type obtained from WSM

(Heidbach et al., 2016) by computing Regime misfit (Figure 5, left panel). The average misfit is lowest for LITHO1 model with

a value of 0.59 (Figure 5g), while CRUST2 model shows the highest average misfit of 0.77 (Figure 5a). High misfits $(2-3)$ are

observed North of MRF and Tehran for CRUST2, while lowest $(< 1)$ in case of LITHO1, suggesting that the dominant mode

of faulting in this area is possibly thrust as opposed to normal deformation predicted by CRUST2. In central Iran, $S_{Hmax}$ misfit

is low $(< 1)$ when the dominant mode of deformation is strike-slip as predicted LITHO1 model.

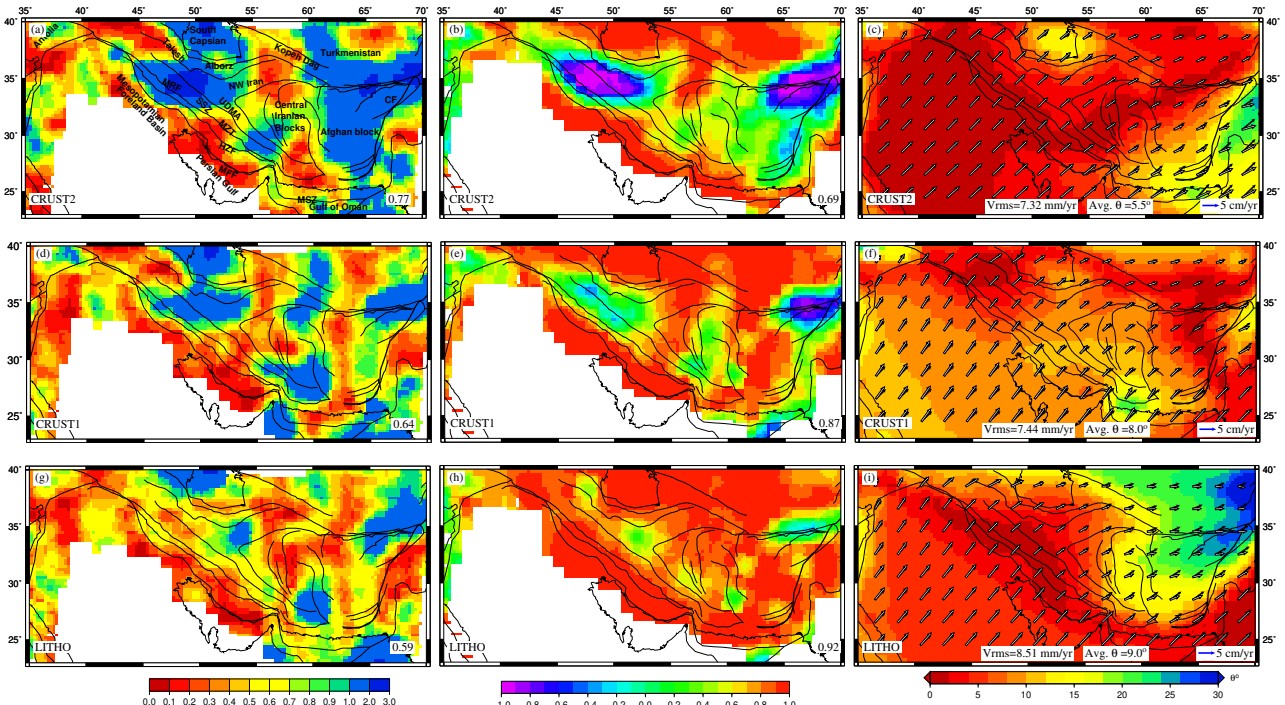

**Figure 5.** (Left Panel) Total misfit between observed and predicted $S_{Hmax}$ from GPE variations. Correlation coefficients between strain rate tensors obtained from Kreemer et al. (2014) and deviatoric stresses predicted using GPE variations are shown in the middle panel, with average regional correlation coefficients on each figure's bottom right. (Right panel) Observed velocities (black) and predicted plate velocities(white) from GPE variations in the NNR frame, plotted on the top of angular misfit between both.

On calculating the correlation between the predicted deviatoric stresses and GSRM strain rates, the LITHO1 model shows

the highest average correlation (0.92) (Figure 5, middle panel). The correlation is found to be extremely poor $(\sim -1)$ for

CRUST2 model in the north of MRF (Figure 5b). Such poor correlation suggests that the predicted stresses differ entirely from

those causing deformation. For example, anti-correlation north of MRF suggests that the dominant mode of deformation in this

area might be thrust rather than normal faulting. Again, the correlation coefficient is less than 0.2 in the central Iranian Block

for CRUST2 and CRUST1 models (Figure 5b,e), while LITHO1 model shows a better correlation suggesting the strike-slip

type of deformation to be more prominent in central Iran (Figure 5h).

We predicted the plate velocities for all three models in the NNR frame and compared them with observed plate velocities obtained from Kreemer et al. (2014) (Figure 5 right panel). CRUST2 gives the least RMS error (7.32 mm/yr) and the lowest angular misfit (5.5°) (Figure 5c). LITHO1 model shows high misfits (> 20°) between observed and predicted velocities in the east of central Iran (i.e. Afghan Block)(Figure 5i). Both CRUST2 and LITHO1 models predict the plate velocities very close to observed ones in the Zagros mountains, as shown by nearly zero angular misfits along Zagros (Figures 5c & i). CRUST1 performs average in predicting the plate velocities in the study area (Figure 5f).

Interestingly, the use of thicker lithosphere to calculate GPE leads to the introduction of more compressional stresses in the region (Figure S2a-f). The average misfit between predicted and observed $S_{Hmax}$ is found to be lowest for the 200 km thick lithosphere (Table S2). Similarly, the correlation between strain rate tensor and predicted stresses,; and rms error between observed and predicted NNR velocities show significant improvement for thicker lithosphere. However, the improvement in fit is better for CRUST2 as opposed to the other two models, CRUST1 and LITHO1, where the misfit between observed and predicted velocities show an increase. Thus, we can say that while considering lithospheric contributions only, the thicker lithosphere does a better job of explaining the observed deformation indicators (Table S2).

## 4.2 Stress and deformation due to Mantle Convection

The deviatoric stresses predicted using all four mantle convection models are found to be mostly compressional along MFF (Figure 6). All models, except for SAW642AN, predict the strike-slip mode of faulting in NW parts of Zagros with nearly E-W oriented extensional axes and N-S compressional axes (Figures 6a,e & g). On the other hand, SAW642AN shows predominant compression within this area (Figure 6e). S40RTS, 3D2018_Sv, and S2.9_S362 show strike-slip deformation in NW parts of SSZ, UDMA and NW Iran. Central Iran is predicted to have mostly compressional stresses by all models except for S40RTS. Thrust type of deformation is predicted in Afghan Block by all models with some intermittent strike-slip deformation. SINGH_SAW model predicts the whole Afghan Block in the strike-slip regime (Figure 6g-h). S40RTS and S2.9_S362 predict higher stress magnitude in NW parts of the Zagros Orogeny system and Central Iran compared to other models.

The misfit between observed and predicted $S_{Hmax}$ is found to be much lower for mantle convection models (0.54-0.57) (Figure 7 left panel), than those of GPE only models (Figure 5 left panel), evidently showing the importance of mantle flow. The lowest average misfit is observed for SAW642AN (0.54) (Figure 7d). Though the misfit increases in the east, Lut block, and near MSZ. The correlation of predicted deviatoric stresses with GSRM strain rates improves over GPE only models (Figure 7 middle panel), with SAW642AN yielding the highest correlation coefficient (0.91) (Figure 7e). Correlation drops below 0.4 parts of central Iran. S40RTS performs predicts the plate velocities closest to the observed one, out of all models, with the least RMS error (~6.20 mm/yr) between predicted and observed plate velocities (Figure 7c). On the other hand, SAW642AN and 3D2018_S40RTS models show high misfits (rms error ~10 mm/yr), as they are unable to match observed plate velocities in Zagros-Iran plateau, both in orientations and magnitude (Figures 7f & i).

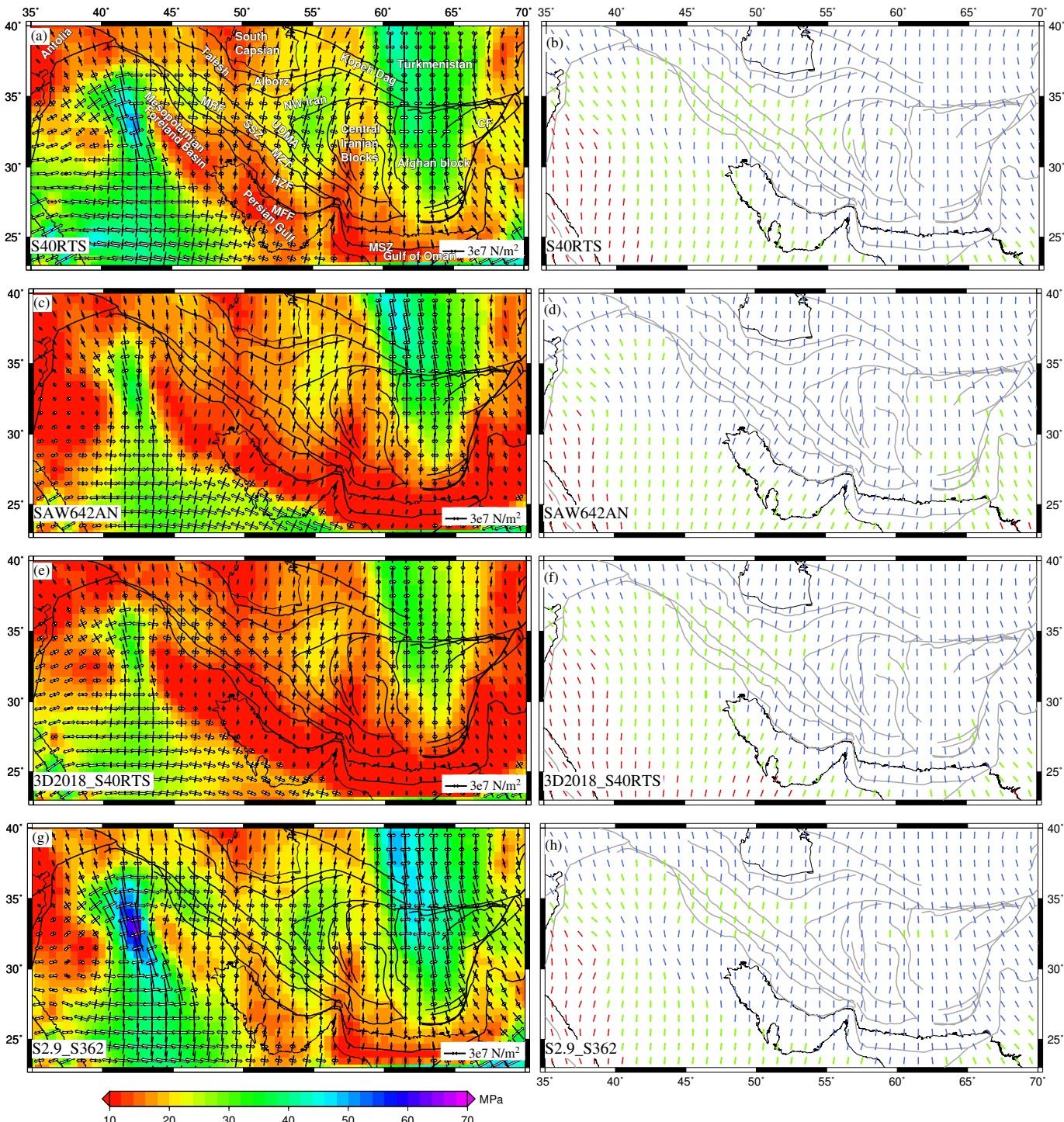

**Figure 6.** (Left Panel) Deviatoric stresses predicted using mantle tractions derived from various tomography models for GHW13 viscosity structure, plotted on the second invariant of deviatoric stresses. The white arrows denote tensional stresses, and the black arrows indicate compressional stresses. $S_{Hmax}$ predicted from mantle tractions is shown in the right panel. Red denotes the tensional regime, blue is for thrust, and green is for the strike-slip regime.

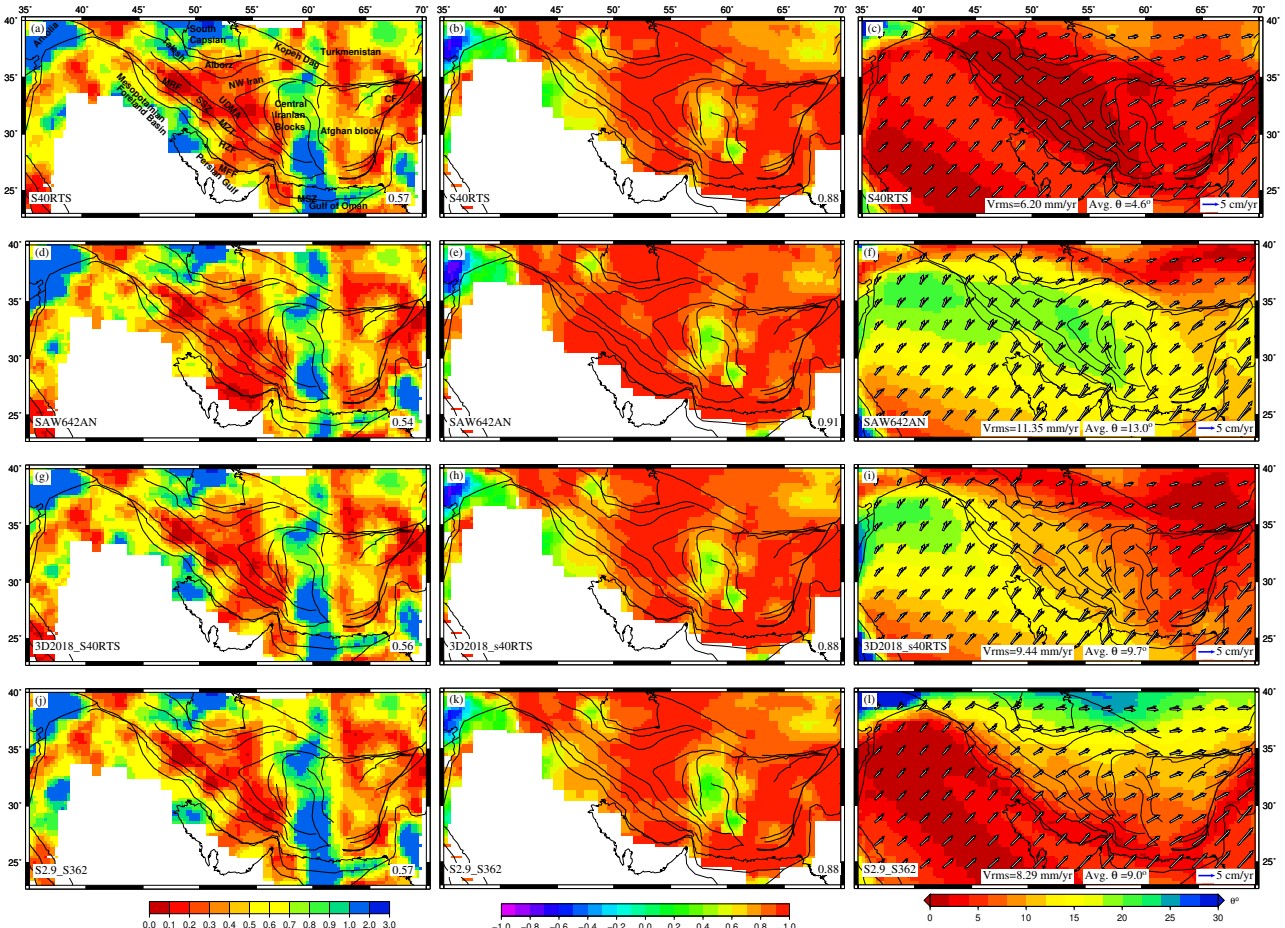

**Figure 7.** Parameters predicted from mantle tractions and their comparisons with observables. (Left panel) Total misfit between $S_{Hmax}$ obtained from WSM (Heidbach et al., 2016) and those predicted using mantle tractions derived from various tomography models using GHW13 viscosity structure. Correlation coefficients between strain rate tensors obtained from Kreemer et al. (2014) and deviatoric stresses predicted using basal tractions are shown in the middle panel, with average regional correlation coefficients on each figure's bottom right. (Right panel) Observed velocities (black) and plate velocities predicted using mantle tractions (white) in the NNR frame plotted on the top of angular deviation between both.

As discussed above, mantle convection models perform better in predicting deviatoric stresses in the study area, which is evident by high correlation between predicted stresses and observed strain rates; and low misfits between observed and predicted $S_{Hmax}$. However, the error in predicting plate velocities is higher for mantle convection models than GPE only models. As there are still significant misfits in fitting the observables, we added the deviatoric stresses predicted from GPE differences and Mantle convection models to constrain the total stress field in the Zagros-Iranian plateau that may account for both forces.

We also ran S40RTS model with LAB (Lithosphere-Asthenosphere boundary) at 150 and 200 km (Figure S2g-h). Similar to GPE models, the fit to observed data shows an improvement when LAB is at 200 km, though the stress patterns do not change significantly (Table S2).

## 4.3 Stress and deformation by GPE and Mantle convection

Adding mantle contributions to GPE only models led to significant changes in total deviatoric stresses for all models (Figure 8,9,10). There is a significant increase in total stress magnitude of the entire study area; except for north of MRF and SE of central Iran, which show slightly lower stresses (< 16 MPa) for combined models of CRUST2 and mantle convection (Figure 8). These models show predominant compression in most of Zagros, SSZ, UDMA, NW and central Iran, except for the strike-slip type of deformation in NW parts. The joint models of CRUST1 and mantle convection predict higher stresses (> 25 MPa) in NW Iran and at MFF (Figure 9). Interestingly, the stresses drop below 20 MPa towards the north of HZF, MRF till the south Capsian. The combined models of CRUST1 and mantle convection show compressional stresses are dominant in the study area, with occasional strike-slip faulting in the north-west (Figure 9 right panel). The stresses predicted by combined models of LITHO1 and mantle convection models are higher in magnitude than other models in the study area (> 25 MPa) (Figure 10). S40RTS+litho and S2.9_S362+litho models show high stresses in Zagros (>50 MPa)(Figure 10a,g).

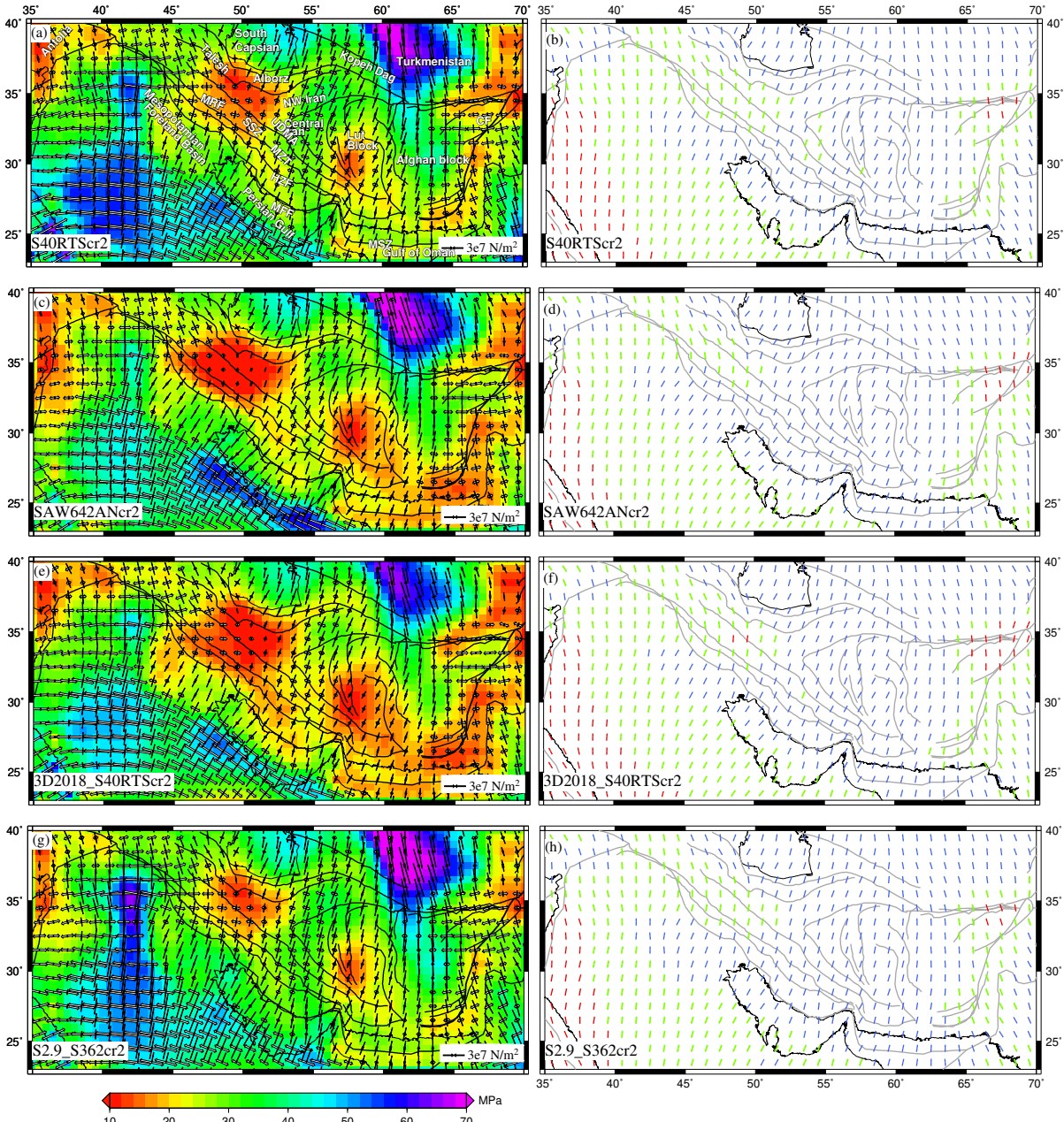

**Figure 8.** (Left panel) Deviatoric stresses predicted using combined effects of GPE computed from CRUST2 and mantle tractions derived from various tomography models plotted on top of their second invariants. The white arrows denote tensional stresses, and the black arrows indicate compressional stresses. The right panel shows $S_{Hmax}$ predicted from these models. The red lines denote the tensional regime, blue is for thrust, and green is for the strike-slip regime.

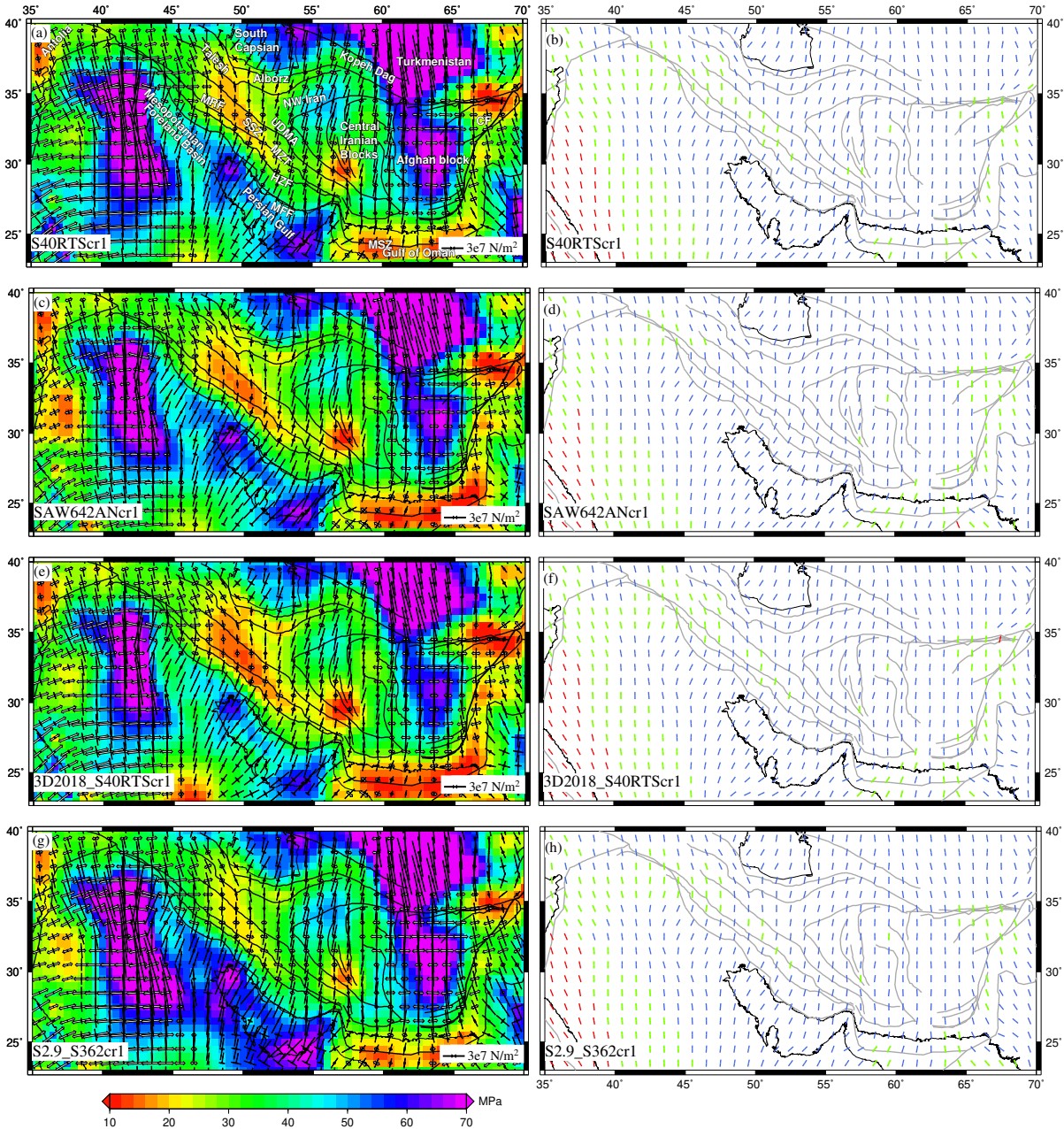

**Figure 9.** (Left panel) Deviatoric stresses (a-d) predicted using combined effects of GPE computed from CRUST1 and mantle tractions derived from various tomography models plotted on top of their second invariants. The white arrows denote tensional stresses, and black arrows indicate compressional stresses. The right panel shows $S_{Hmax}$ predicted from these models. The red lines denote tensional regime, blue is for thrust and green is for strike-slip regime.

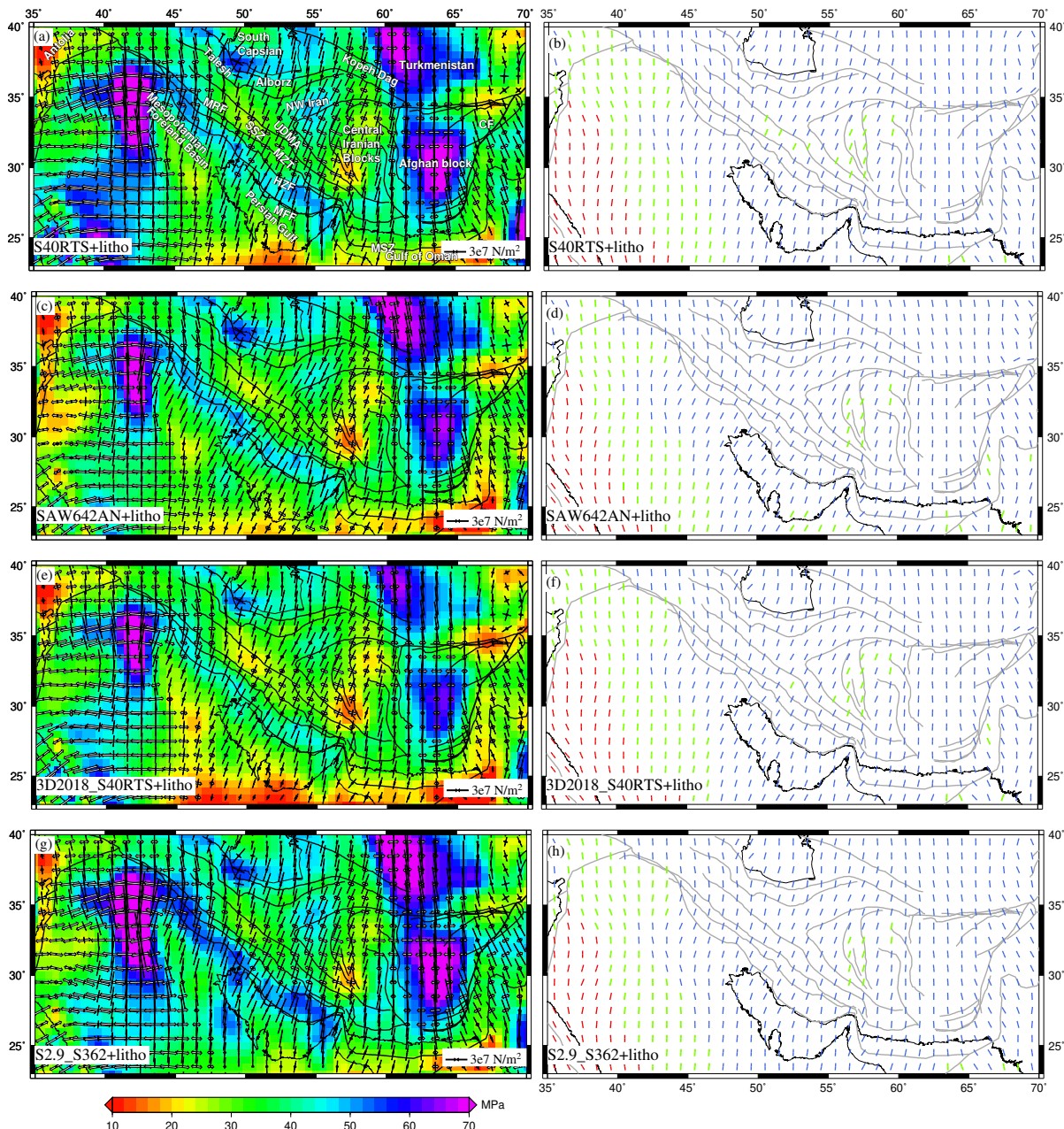

**Figure 10.** (Left panel) Deviatoric stresses (a-d) predicted using combined effects of GPE computed from LITHO and mantle tractions derived from various tomography models plotted on top of their second invariants. The white arrows denote tensional stresses, and the black arrows indicate compressional stresses. The right panel shows $S_{Hmax}$ predicted from these models. The red lines denote tensional regime, blue is for thrust, and green is for the strike-slip regime.

The combined models show a lower misfit between observed and predicted $S_{Hmax}$ (Figure 11), especially when compared to GPE only models (Figure 5 left panel). SAW642AN+litho showed the lowest average misfit of 0.47 (Figure 11f). Interestingly,

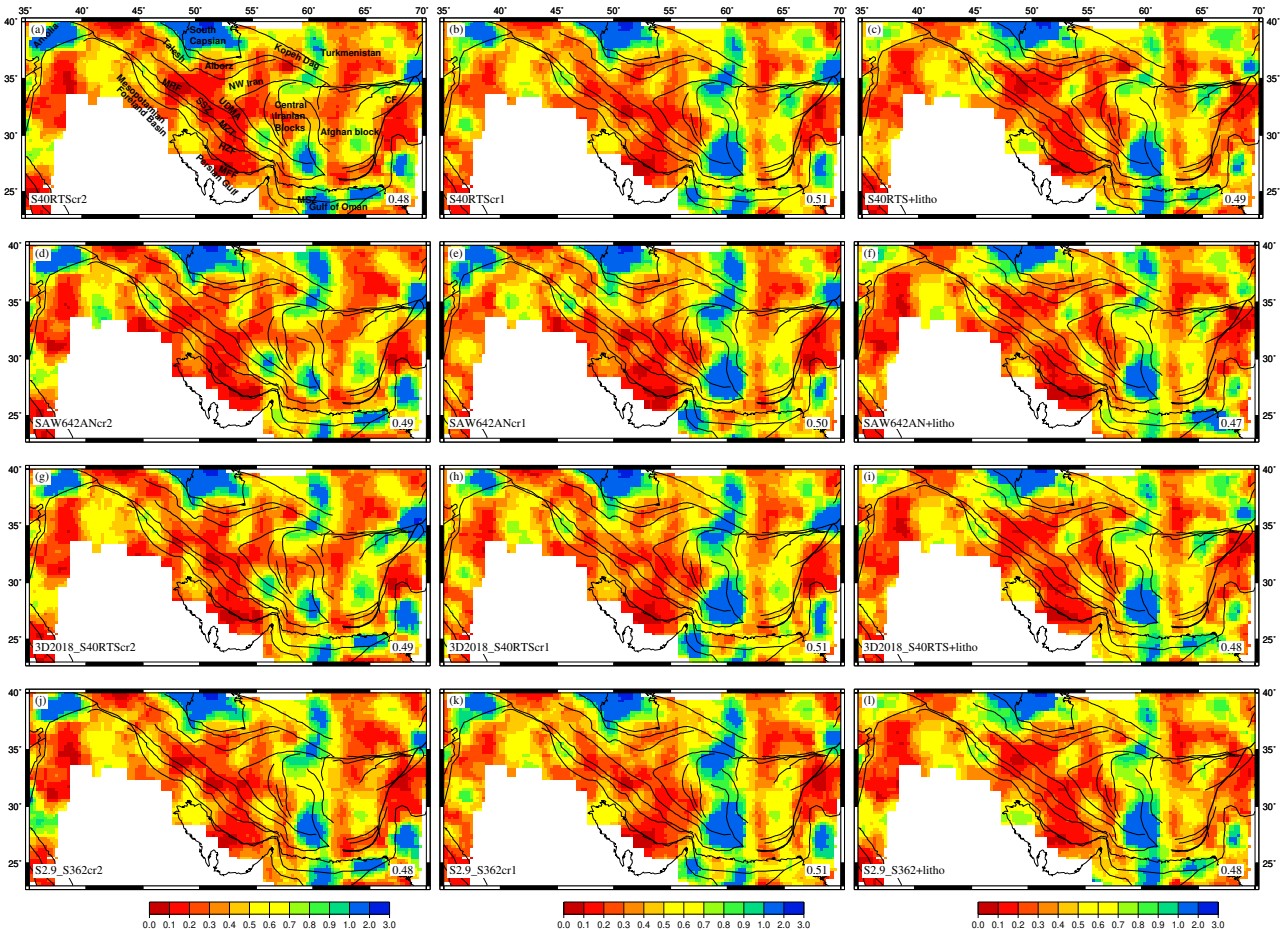

**Figure 11.** Total misfit between observed $S_{Hmax}$ from WSM (Heidbach et al., 2016) and $S_{Hmax}$ predicted using combined effects of GPE computed from different crustal models and mantle tractions derived from various tomography models.

SAW642ANcr2 and 3D2018_S40RTScr2 show low misfits in the Zagros-Iranian plateau region, despite not having the lowest average misfit (Figures 11d & g). The higher misfits in NW Iran and SE of the central Iran block observed for GPE only models get reduced significantly due to the addition of mantle derived stresses, referring to the importance of mantle convection in these areas.

As we look at the correlation between predicted stress tensors and GSRM strain rate tensors, the overall correlation is better for combined models (Figure 12), especially for combined models of LITHO1 and mantle convection (Figure 12 right panel). A high average correlation coefficient of 0.94 is observed for SAW642AN+litho, 3D2018_S40RTS+litho as well as S2.9_S362+LITHO1 (Figures 12f, i & l). Despite an overall improvement in correlation between observed strain tensors and predicted deviatoric stresses, the correlation is found to be much poor in areas such as NW parts of Zagros and east of central Iranian block, for combined models of mantle convection and GPE only models of CRUST2 & CRUST1 (Figure 12 left and

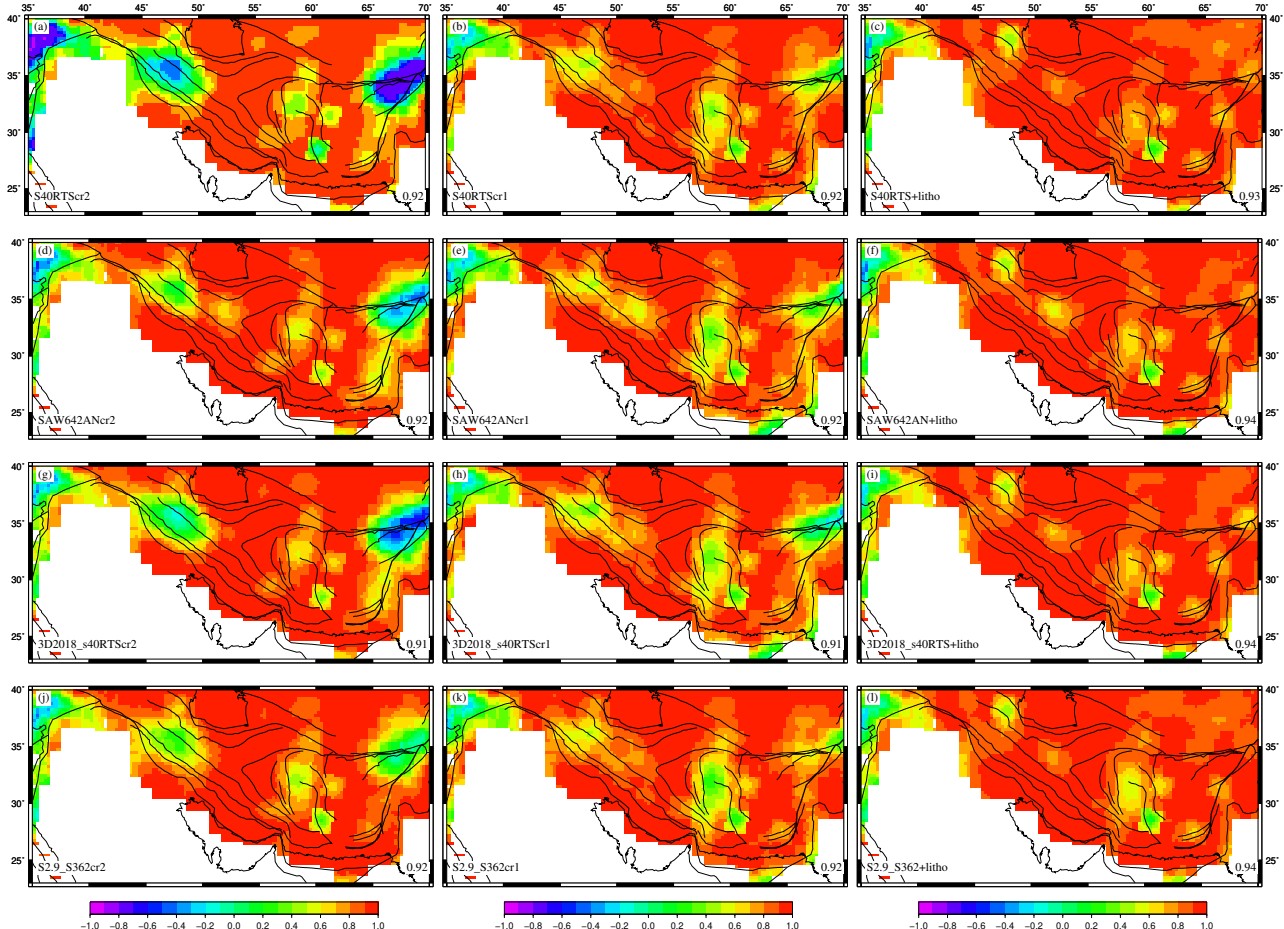

**Figure 12.** Correlation coefficients between strain rate tensors from Kreemer et al. (2014) and deviatoric stress tensors predicted using combined effects of GPE computed from different crustal models and mantle tractions derived from various tomography models. The figure shows the average correlation coefficient in the lower right corner of the figure.

middle panels). In NW Zagros, mantle only models are found to perform much better, as they show better correlation (Figure 7 middle panel), thus suggesting mantle derived stresses are needed to be much higher than those from GPE to explain the observed deformation in these areas.

     Again the combined models of GPE and mantle tractions give lower rms errors, when predicted plate velocities are compared to the observed ones. S40RTScr2 shows the least rms error (3.28 mm/yr) and the least average angular misfit (3.0°) between
predicted and observed plate velocities (Figure 13a). Relatively the combined models of S40RTS/S2.9_S362 and GPE perform much better than other models in predicting the orientation and magnitudes of plate velocities. Significant misfits are observed for SAW642ANcr1 and 3D2018_S40RTScr1 models. The joint models of S40RTS and GPE for thicker lithosphere do not offer any significant changes in stresses and their fit to observed data (Table S2) (Ghosh et al., 2009; Jay et al., 2018; Hirschberg et al., 2018). Thus, considering the lithosphere base at 100 km appears to be a satisfactory approach.

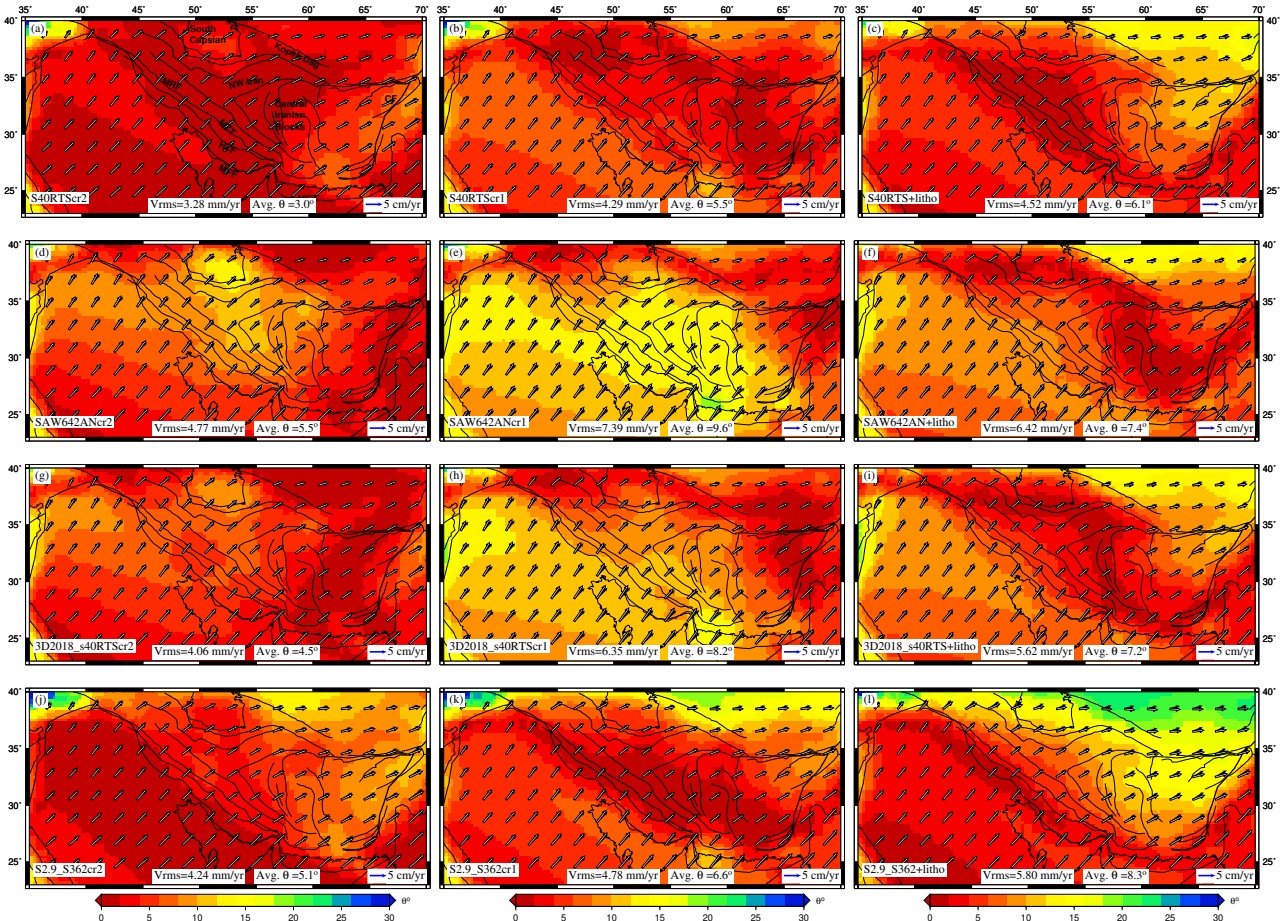

**Figure 13.** Plate velocities predicted using combined effects of GPE computed from different crustal models and mantle tractions derived from various tomography models plotted on top of angular misfit (θ). Black arrows represent observed NNR velocities (Kreemer et al., 2014) and white ones denote predicted velocities.

## 5    Discussion

The Zagros-Iranian plateau region is formed due to the convergence of the Arabian plate towards the Eurasian plate. Zagros mountain belt demarcates the southwestern boundary of the deformation zone, whereas it is bounded by the Makran subduction zone in the southeast and by Afghan Block in the east. Kopet-Dagh and Arborz act as this region's northeastern and northern boundaries (Irandoust et al., 2022). We modeled the stresses and deformation parameters in the study area by solving the force balance equation using the finite element method for a global grid of $1° \times 1°$ resolutions, considering two primary sources of stresses; GPE and mantle tractions. GPE was calculated using the thickness and density variation from the different global models like CRUST1.0, CRUST2.0 and LITHO1.0. The shear tractions were computed from density derived mantle convection model.

The magnitude of stresses due to GPE variations was below 15 MPa in the Iranian plateau for CRUST2 and CRUST1 models (Figures 4a & c). However, LITHO1 model predicted higher stresses ($> 30$ MPa) with predominant compression in parts of the Zagros-Iran region and Afghan block. Most of the convergence of Arabian and Eurasian plates has been accommodated through shortening across Zagros (Irandoust et al., 2022; Khodaverdian et al., 2015). Walpersdorf et al. (2006); Hessami et al. (2006) suggested nearly pure N-S shortening of $8 \pm 2$ mm/yr in southeastern Zagros. The convergence occurs perpendicular to the simply folded mountains and is restricted to the shore of Persian Gulf. Earthquake focal mechanisms also show reverse faulting within this area (Berberian, 1995; Hatzfeld et al., 2010; Hatzfeld and Molnar, 2010; Irandoust et al., 2022). In our study, LITHO1 model predicted thrust mode of faulting within Zagros, which is consistent with these results. In NW Zagros, Hatzfeld et al. (2010); Hatzfeld and Molnar (2010); Jackson and McKenzie (1984); Khorrami et al. (2019); Talebian and Jackson (2002) and various others have suggested partitioning of deformation. The oblique shortening is partitioned into strike-slip faulting that is accommodated by MRF, while shortening occurs perpendicular to the mountain belt (Hatzfeld et al., 2010; Hatzfeld and Molnar, 2010; Jackson and McKenzie, 1984; Khorrami et al., 2019; Talebian and Jackson, 2002). On considering lithospheric models only, we predicted the normal mode of faulting to be dominant in this area for CRUST2. On the other hand, CRUST1 model predicted strike-slip components in the northern segment of MRF, while LITHO1 showed thrust type of deformation in this area. Interestingly, the misfits of predicted parameters with various observations of $S_{Hmax}$, strain rates and plate velocities were found to be lowest for LITHO1 model, thus arguing for thrust type of deformation in this area. SSZ in the north of MZT consists of various thrust systems (Alavi, 1994). CRUST1 predicted thrust mode of faulting in this region, while CRUST2 and LITHO1 models showed intermittent strike-slip type of faulting. Alborz as well as Kopeh Dagh in the north has also been subjected to reverse faulting (Allen et al., 2003; Hatzfeld and Molnar, 2010; Hollingsworth et al., 2010; Irandoust et al., 2022; Khodaverdian et al., 2015), which has also been shown by CRUST1 and LITHO1 models. Models predicting thrust in Talesh mountains showed low misfits to observation suggesting thrusting of the mountain range over the basin with slip vectors directed towards the South Caspian Sea (Irandoust et al., 2022). The N-S convergence in Kopeh-Dagh range was predicted by LITHO1 model considering the contribution from lithospheric density and topographic variations only. The shearing between Central Iran and Afghan Block caused due to varying rates of shortening across the Zagros, Alborz and Caucasus, is accommodated by strike-slip faults near Lut block boundaries (Khorrami et al., 2019; Vernant et al., 2004; Walpersdorf et al., 2014). Again, LITHO1 model predicted similar strike-slip deformation in these areas; however, CRUST2 and CRUST1 failed to do so.

The stresses predicted using basal tractions were mostly compressional in southeastern Zagros owing to the convergence of Arabia-Eurasia (Figure 6). However, all models, except SAW642AN predicted strike-slip type of deformation in the northwestern Zagros (MRF), which concurs with the results from various studies (Hatzfeld et al., 2010; Hatzfeld and Molnar, 2010; Jackson and McKenzie, 1984; Khorrami et al., 2019; Talebian and Jackson, 2002). The mantle derived stress parameters showed a better fit to observables than those from GPE variations (Figures 7 left and middle panel), though the correlation dropped below 0.5 in Central Iran. Here, mantle convection models found compressional type of deformation, while Baniadam et al. (2019); Khorrami et al. (2019) suggested that strike-slip faulting along the fault system bounding Lut Block. The velocity misfits were very high for all models except S40RTS (Figure 7 right panel). Although we used four tomography models to

compute the mantle-derived stresses, the stress regimes for all models are found to be similar, with varying magnitudes. Such results suggest that nearly all four seismic tomography models are relatively consistent in predicting the stresses in this region.

Adding the GPE derived stresses to those from the mantle to obtain the total lithospheric stress field showed a notable improvement in constraining the observed deformation parameters. The final stress regimes also varied significantly depending on particular combinations of GPE and mantle convection models. All joint models of CRUST2 and mantle tractions showed lower magnitudes of stresses ($< 15$ MPa) in the north of MRF, Tehran and southern Lut block. The stresses showed an obvious increase in these areas for other models. Significantly higher stresses ($> 30$ MPa) were also observed near the collisional front (MFF) for all models. On comparing with observations, combined models of CRUST2 and mantle tractions showed significant improvement in fit, except in areas north of MRF and Tehran. CRUST1 model when added with mantle contribution, predicted thrust faulting along the faults bounding Lut Block, leading to poor correlation ($< 0.5$). On the other hand, combined LITHO1 and mantle convection models gave a much better fit in this area, as they predicted strike-slip faulting. The use of different mantle convection models is much less sensitive in the Iran-Zagros region, as most models can match various surface observables reasonably well.

On running various models and comparing the stresses in Zagros-Iran, we try to explain the relative roles of GPE and mantle tractions in causing observed deformation. The contributions from both sources vary significantly among different models. However, these variations arise mainly from GPE only models, which may be due to uncertainties in crustal models of this area. Another interesting observation from this study is that the role of GPE in the study region may not be that significant, as mantle derived stresses were able to explain many of the deformation indicators. To get a quantitative constraint on the best model, we computed a total error as given below:

$$Total\, error = S_{Hmax}\, error + 1 - C_{strain} + V_{rms} \tag{7}$$

$S_{Hmax}$ error in the above equation is calculated as mentioned in section 3.4, while $C_{strain}$ is the correlation computed using equation 6. $V_{rms}$ is the rms error between predicted and observed velocities. The total errors calculated using equation 7 have been tabulated in Table 1. S40RTScr2 is found to have the lowest error.

We also calculated plate velocities with respect to the Eurasian plate (Figure 14) and compared them with observed GPS velocities relative to Eurasia. The GPS velocities were obtained from various studies conducted in this area (ArRajehi et al., 2010; Bayer et al., 2006; Frohling and Szeliga, 2016; Khorrami et al., 2019; Masson et al., 2006, 2007; Raeesi et al., 2017; Reilinger and McClusky, 2011; Vernant et al., 2004). GPS measurements show a northward convergence rate of $\sim 22$ mm/yr for Arabia relative to Eurasia (Reilinger et al., 2006; Vernant et al., 2004), however, it varies significantly along the Zagros. The southeastern Zagros show the highest convergence rates of $\sim 25$ mm/yr oriented in the north-northeast directions. GPS vectors are oriented northward in Central Zagros, which transitions north-northwest in NW parts of Zagros with the lowest convergence rates of $\sim 18$ mm/yr (Hatzfeld and Molnar, 2010; Hatzfeld et al., 2010; Khorrami et al., 2019). Vernant et al. (2004) suggested that MSZ accommodates most of the shortening ($19.5 \pm 2$ mm/yr) in the east of $58°E$, while fold and thrust belts of Zagros, Alborz and Caucasus collectively accommodate the shortening in west of $58°E$. GPS velocities in the east

**Table 1.** Summary of quantitative comparison of predicted results of various models with observed data.

| Model | $S_{Hmax}$ misfit | Strain rate correlation | RMS error (mm/yr) | Angular misfit | Total error |
|---|---|---|---|---|---|
| CRUST2 | 0.77 | 0.69 | 7.32 | 5.5 | 3.07 |
| CRUST1 | 0.64 | 0.87 | 7.44 | 8 | 2.78 |
| LITHO1 | 0.59 | 0.92 | 8.51 | 9 | 2.81 |
| | | | | | |
| S40RTS | 0.57 | 0.88 | 6.2 | 4.6 | 2.51 |
| SAW642AN | 0.54 | 0.91 | 11.35 | 13 | 3.06 |
| 3D2018_S40RTS | 0.56 | 0.88 | 9.44 | 9.7 | 2.92 |
| S2.9_S362 | 0.57 | 0.88 | 8.29 | 9 | 2.81 |
| | | | | | |
| S40RTScr2 | 0.48 | 0.92 | 3.28 | 3 | 1.75 |
| SAW642ANcr2 | 0.49 | 0.92 | 4.77 | 5.5 | 2.13 |
| 3D2018_S40RTScr2 | 0.49 | 0.91 | 4.06 | 4.5 | 1.98 |
| S2.9_S362cr2 | 0.48 | 0.92 | 4.24 | 5.1 | 2.00 |
| | | | | | |
| S40RTScr1 | 0.51 | 0.92 | 4.29 | 5.5 | 2.05 |
| SAW642ANcr1 | 0.5 | 0.92 | 7.39 | 9.6 | 2.58 |
| 3D2018_S40RTScr1 | 0.51 | 0.91 | 6.35 | 8.2 | 2.45 |
| S2.9_S362cr1 | 0.51 | 0.92 | 4.78 | 6.6 | 2.15 |
| | | | | | |
| S40RTS+litho1 | 0.49 | 0.93 | 4.52 | 6.1 | 2.07 |
| SAW642AN+litho1 | 0.47 | 0.94 | 6.42 | 7.4 | 2.39 |
| 3D2018_S40RTS+litho1 | 0.48 | 0.94 | 5.62 | 7.2 | 2.27 |
| S2.9_S362+litho1 | 0.48 | 0.94 | 5.8 | 8.3 | 2.30 |

of Iran (Afghan Block) are very small in magnitude. To the west, velocities increase showing westward rotation of Anatolia (Khorrami et al., 2019; Reilinger et al., 2006). The northern part of Iran shows that GPS vectors are aligned towards the northeast. We found that the combined model of S40RTS and CRUST2 can approximately match the GPS velocities (Figure 14a). Predicted plate velocities with respect to the fixed Eurasian plate show a northward movement of $\sim 2-3$ cm/yr in the southeastern Zagros. The plate moves in NNE direction east of central Zagros ($53°$ E). On the other hand, west of $53°$ E shows a movement in NNW direction, becoming much more prominent in the north. However, the convergence rates in the east of Iran i.e. Lut Block as well as Afghan Block, is predicted to be much higher ($\sim 1-2 cm/yr$) than those suggested by various observations. Plate velocities predicted by joint models, S40RTScr1 and S40RTS+LITHO1 show nearly N-S contraction of very high magnitudes (4-5 cm/yr) throughout the region (Figure S3), which suggests much higher rates of deformation than those suggested by above-mentioned studies.

We also used shear wave splitting measurements to further study the deformation in the Zagros-Iran region by comparing them with $S_{Hmax}$ (Figure 14b). The fast polarization directions (FPDs) are the indicators of seismic anisotropy. We consider two primary causes of seismic anisotropy; induced by stress and due to the structure of the region (Yang et al., 2018). If the FPDs are parallel to $S_{Hmax}$ orientations, it suggests that anisotropy is associated with stress. On the other hand, the latter kind

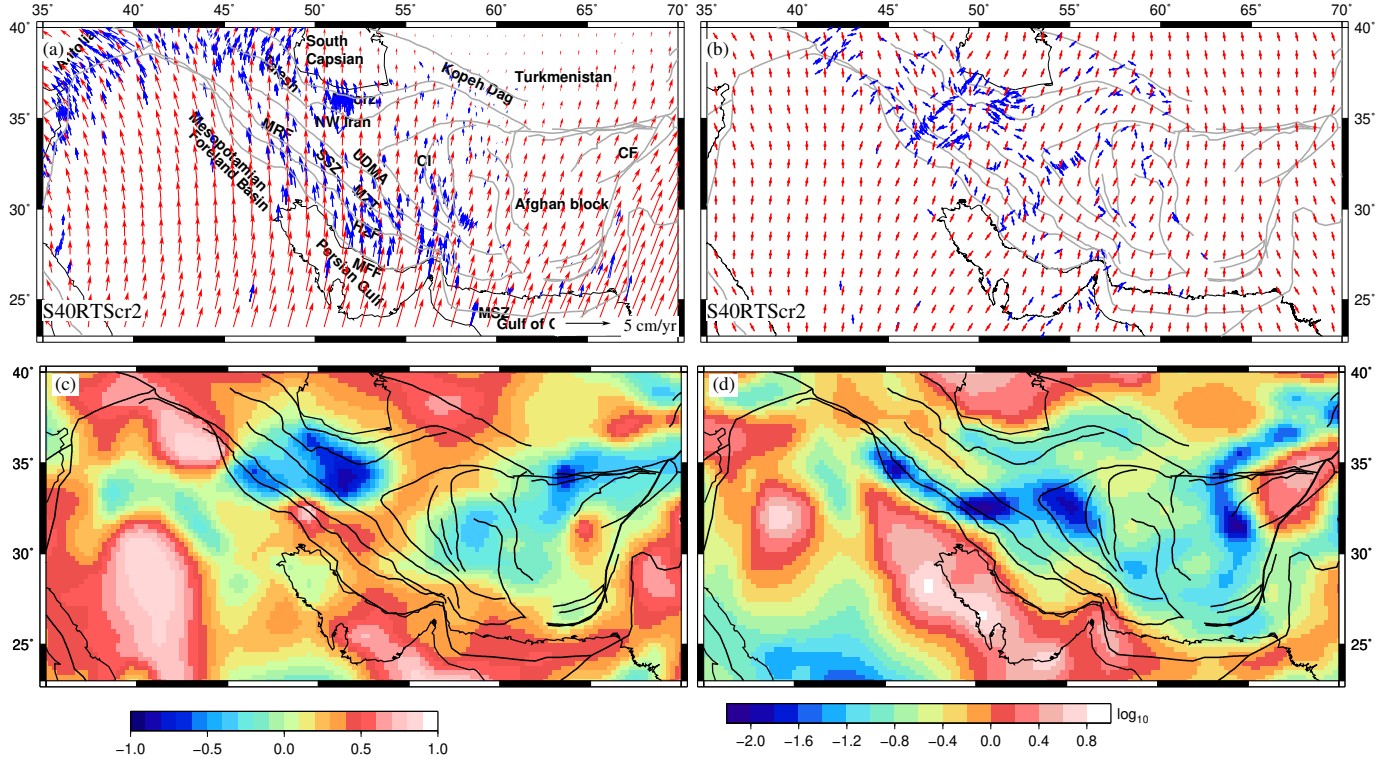

**Figure 14.** Predicted parameters of the best fit model, S40RTScr2. (a) GPS (blue) and predicted (red) plate velocities with respect to a fixed Eurasian plate, (b) FPDs (blue) and $S_{Hmax}$ (red) are plotted for the best fit model, (c) Correlation between deviatoric stresses predicted from GPE and mantle convection models, and (d) ratio $(T_1/T_2)$ of the second invariant of deviatoric stresses from GPE $(T_1)$ to those from mantle tractions $(T_2)$.

of anisotropy is related with the alignment of fault, fast axes of minerals that may cause polarization, and sedimentary bedding planes. The FPDs in our study were obtained from Sadeghi-Bagherabadi et al. (2018); Kaviani et al. (2009, 2021). The FPDs are subparallel to $S_{Hmax}$ orientations in NW Zagros, Arabian plate, northern Iran and MSZ. Such a correlation between both indicates that anisotropy in this region may be stress induced. Additionally, the correlation of $S_{Hmax}$ orientations and FPDs

argues for a good coupling between lithosphere and mantle in those areas. In contrast, Sadeghi-Bagherabadi et al. (2018) showed FPDs parallel to the strike of the fault (sub-parallel to $S_{Hmax}$ directions of CRUST2), suggesting seismic anisotropy mainly reflects the deformation in the lithospheric mantle. Again, FPDs are subparallel to the strike of range in the northeastern Iran, eastern Kopeh Dagh and central Alborz indicating structure-induced anisotropy caused by strong shearing along the strike-slip faults (Gao et al., 2022; Kaviani et al., 2021).

To explore the relative roles of lithospheric and mantle derived stresses, we compared the deviatoric stresses from CRUST2 to those from S40RTS. We performed a correlation between both stresses by using equation 5 and found a high correlation ($> 0.5$) near MSZ and central Zagros (Figure 14c). The correlation degrades north of the simply folded mountains and NW Iran. The stresses are anti-correlated in northwestern parts of higher Zagros, north of MRF and Tehran, as CRUST2 predicted

NNE-SSW tension (Figure 4b) as opposed to the strike-slip faulting predicted by S40RTS (Figure 6b). Lut Block also shows a slight anticorrelation between stresses ($\sim -0.5$), as the stresses predicted by CRUST2 are very low. The log of the ratio of second invariants of deviatoric stresses from GPE variations ($T_1$) to that of mantle tractions ($T_2$) is plotted in Figure 14d. Positive values of logarithmic ratio suggest the dominance of GPE derived stresses over mantle ones, as observed in the south of the collisional boundary (MFF). The ratio is negative in most parts of the Iranian plateau and Zagros, indicating that the magnitude of mantle derived stresses is higher than that of GPE, especially in higher Zagros and central Iran (Figure 14d).

The deformation in the Zagros-Iran plateau region has been found to exhibit various similarities to another similar complex collision zone, i.e. the Himalaya-Tibetan plateau region as both continental collisions went through many of the same processes. The high topography in both collisions reflects ongoing crustal deformation through crustal thickening and shortening. However, there are differences in the convergence rates, total amounts of convergence and various stages of development of the Zagros-Iran and Himalaya-Tibet regions (Hatzfeld and Molnar, 2010). Singh and Ghosh (2020) studied the deformation in the Himalaya-Tibet region by joint modeling of lithosphere and mantle. They showed that GPE plays a crucial role in the ongoing deformation of the India-Eurasia collision zone as it leads to the observed E-W extension in the Tibetan plateau. In contrast, we found that GPE has a much lesser role in the Zagros-Iran plateau region (Figure 14d), and no normal mode of faulting is observed in this area. In the Zagros-Iran plateau region, mantle convection appears to be the primary driver of deformation in most parts as discussed above. Despite these differences, numerical models argue for a good coupling between the lithosphere and mantle in both collision zones, which is also supported by seismic anisotropy studies in both regions (Kaviani et al., 2021; Singh et al., 2016; Sol et al., 2007).

## 6 Conclusion

The Zagros-Iranian plateau region has large deformations along and across the collision zones. Therefore, we conducted numerical simulation studies for stress and deformations. The stresses predicted in this region were primarily compressional, with magnitudes lower than 30 MPa. The southeastern boundary of Zagros was found to be under high stress, which is also reflected by higher convergence rates. Mantle convection models were able to constrain most observations in the Iranian plateau. However, the misfits with observations were much larger in the east of Iran, when only mantle contributions were considered. The combined models of lithosphere and mantle-derived stresses can explain surface observables in most of the area, suggesting a good lithosphere-mantle coupling, except for east of Iran. The shearing in those areas was predicted by lithosphere models, though variation in lithospheric and density structure given by these models lead to varying degree of misfits. Hence, there is a need for better constraint on lithospheric structure in this area.

The mantle derived stresses were found to be much higher than lithospheric stresses, thus the overall stress regimes predicted by combined models were more biased towards the compressional type of stresses. This caused our combined models to predict thrust mode of faulting in most cases, especially when lithospheric stresses were computed from CRUST1 and LITHO1 models. CRUST2 model predicted more extensional stress in the Iranian plateau, which in turn balanced the effect of compressional stresses predicted by mantle convection models; hence leading to prominence of strike-slip mode of faulting in the northwestern

parts of study region. The rate of convergence of Arabia relative to a fixed Eurasia was found to vary along the Zagros orogeny in a similar way to GPS measurements.

**Author Contribution**

S.Singh ran the numerical models for computing various parameters in this manuscript. Both authors, S. Singh and R. Yadav were involved in interpretation of the predicted results. This manuscript was prepared by S.Singh under the supervision of R. Yadav.

*Acknowledgements.* We thank Dr. Attreyee Ghosh for sharing the codes for finite element modeling of stresses. We are also thankful to the director of CSIR-NGRI for permission to publish this work (Ref. No. NGRI/Lib/2022/ Pub-110). This article is published as a part of the
435 MLP-FBR-003(AM) Project. The figures in this article are generated from the open-source GMT modules.

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
