# Peer review of "Numerical modeling of stresses and deformation in the Zagros-Iranian plateau region"

_EGUsphere, 2023_

## Author Response (AR1)

**Comment by Editor**

Dear authors,

I have received two reviews of this manuscript and have closed the discussion.

As you will see, while reviewer 2 is rather positive, reviewer 1 has raised some important concerns that need to be addressed for the manuscript to be considered for publication. I would particularly point out to the requested tests of different LAB models, and a significant improvement in the text and structure. Do not feel, in any way, obliged to cite the references indicated by reviewer 2.

All the best,

Simone.

**Reply to Editor**

We thank Dr Simone Pilia (Editor) for her comments and suggestions to improve the quality of the manuscript. We have incorporated the suggestion given by the two reviewers.

**RC1: 'Comment on egusphere-2023-250', Frédéric Mouthereau**

In this study, the authors test different tomographic models to calculate the impacts of different convection patterns in combination with the GPE arising from different crustal architectures to derive the stresses (SHmax, deviatoric stress orientation, and magnitude) and velocity field across the Zagros-Iran Plateau. They take the same approach as Ghosh et al. (2013). While the approach may seem valid and has never been attempted in the region, the manuscript and science need to be rewritten and rethought in a more meaningful way. My main scientific concern is with the choice of parameters tested. The authors use different tomographic models that are globally consistent and different crustal models that also show little variations. Thus, as expected, the differences between the models tested are only subtle and thus do not provide much information about the dynamics of the region. The varying thickness of the crust is expected to have a much smaller effect on the calculation of tractions than the thickness, density, and viscosity of the lithosphere. However, the authors prescribe a constant lithosphere thickness of 100 km. This essential parameter must be taken into account in the modeling, as well as the lateral variations in density and viscosity between the Arabian Shield and the Zagros, below which several authors have suggested very different lithosphere thicknesses, up to 200 km. This is also true for the Iranian plateau, which is much thinner. In practice, the authors present a short sentence indicating that they use the viscosities of Ghosh et al. 2013, but without further explanation or justification. Variations in LAB need to be tested.

But the main problem is probably the text itself. The introduction is terrible and demonstrates the poor knowledge of the geology of the region. It should be completely reworked. The study should be better justified. What is the key issue addressed? In my opinion, the implementation of lithospheric structures in the model and in the question would be very useful for the readership. The paper is far too short and lacks an explanation for readers to understand what is strong and weak in the previous works and the added value of the present study. Currently, it reads as a copy and paste of the Ghosh et al. 2013 approach applied to the Zagros, but without a minimum explanation and without a full study of the critical parameters. I attach my comments in a pdf file.

Reply

We thank the reviewer for his comments and suggestions to improve the manuscript. We have kept the base of the thin sheet model at 100 km, what is considered to be the average depth of the lithosphere. The reviewer is correct in pointing out that the lithosphere thickness varies from Arabian shield to Zagros to the Iranian plateau. However, for a thin sheet model, we can only integrate up to a constant base and we would require full 3-D lithosphere models in order to address this point in a completely satisfactory manner. So, we tested the model for lithospheric depth of 150 km and 200 km, which we have included in the supplements of the paper. We didn't observe significant differences in style of the deviatoric stresses predicted for the thicker lithosphere with a similar density as the ambient mantle, which is similar to previous studies (e.g. Ghosh et al.,2009; Jay et al., 2018; Hirschberg et al., 2018) showing that the GPE differences and the associated deviatoric stresses do not vary significantly with different depths of integration. Regarding the density variations, Figure 2 of the paper shows the lateral variations in density of the different layers for the study region, which are the input in Finite Element (FE) model itself. We have shown the lateral viscosities used in our model in Figure S1, which has stiff/viscous Arabian shield that the Zagros-Iranian plateau region. We also tested the effect of LAB depth on one of the best fit models (Figure S2).

We incorporated the suggestion in the manuscript. We have given point-by-point answers below.

Line 39 "from the"

Reply: - We have added "from the" as suggested by reviewer (Line 45).

Line 41-42 Replace "Though there has been an increase in the influx of various studies trying to constrain" by "Despite the first-order characteristics of".

Reply: - We have replaced the text (Lines 49-50).

Line 42 "deformation and" → "deformation and present-day"

Reply: - We have added "present-day" in the text (Line 50).

Line 44 "there are debates about various processes in this region, e.g." → "is relatively well understood"

Reply: - We have modified the sentence (Lines 50-51).

Line 44 "timing of collision" → "the timing of collision is debated".

Reply: - We have modified the sentence as per reviewer suggestions (Lines 52-53).

Line 45 Collision culminated in the Late Eocene to early Oligocene. you can add Koshnaw et al. (2018) and Agard et al. (2011).

Reply: - I have modified the sentence and added the references (Line 54-59).

Line 45 Mouthereau et al., 2012

Mouthereau, F., Lacombe, O., Vergés, J., 2012. Building the Zagros collisional orogen: Timing, strain distribution and the dynamics of Arabia/Eurasia plate convergence. Tectonophysics 532–535, 27–60. https://doi.org/10.1016/j.tecto.2012.01.022

Reply: - We have added he reference (Line 58).

Line 46 Remove "have suggested that collision onset time to be in Late Eocene to Oligocene;"

Reply: - We have removed the sentence (Lines 54).

Line 46-47 No. The age of the collision is now well constrained to be between late Eocene-Oligocene but certainly not Paleocene. The works you are citing do not focus on the collision initiation but rather on the evolution of Cenozoic magmatism and magnetic fabrices which do not allow drawing conclusions on the age of the collision.

Reply: - We have modified the sentence (lines 54-59).

Line 48-50 "The Zagros and its foreland area have a great source of natural resources like petroleum. The study area consists of the ocean-continent subduction as well as the continental collisions. The convergence rate of the Arabian plate relative to the Eurasia varies from east to west."

What do you want to say here ? remove or rewrite. These sentence read read like a suite of aimless statements.

Reply: - We have removed the sentence (lines 60-63).

Line 50-51 "These complex structures and convergence velocity variation made the variable tectonic stress and deformation."

So what?

Reply: - We have modified the entire sentence." (Lines 66-76)

Line 52 "seismic" → "seismically"

Reply: - We have modified the text.

Line 52 "earthquake data,"

Sure. Earthquakes indicate it is active but geodesy also. Rephrase.

Reply: - We have rephrased the sentence (lines 73-76).

Line 53-54 " Therefore, the world stress map" → "The world stress map"

Reply: - We have modified the sentence (Lines 82-84).

Line 56-59 "However, the in-situ stress (WSM) and GPS velocity data (ArRajehi et al., 2010; Bayer et al., 2006; Frohling & Szeliga, 2016; Khorrami et al., 2019; Masson et al., 2006, 2007; Raeesi et al., 2017; Reilinger & McClusky, 2011; Vernant et al., 2004) are very limited and sparsely distributed in this region; therefore, there is need for a numerical simulation study to comprehend the knowledge."

The wording is very bad. This part is relevant for the paper's goal but the justification is not well presented. I think you study present useful results do better understand the relative role of GPE and lithosphere structure coupled to mantle convection on the stress orientation. Such a study has never tempted in the Zagros so say it. You may find some elements for introducing your work in Mouthereau et al. (2021).

Mouthereau, F., Angrand, P., Jourdon, A., Ternois, S., Fillon, C., Calassou, S., Chevrot, S., Ford, M., Jolivet, L., Manatschal, G., Masini, E., Thinon, I., Vidal, O., Baudin, T., 2021. Cenozoic mountain building and topographic evolution in Western Europe: impact of billions of years of lithosphere evolution and plate kinematics. Bsgf - Earth Sci Bulletin 192, 56. https://doi.org/10.1051/bsgf/2021040

Reply: - We have rephrased the sentence for a better reading as below (lines 84-86):

"However, in-situ stress data are sparsely distributed and limited, so numerical modeling plays an important role in understanding the kinematics and dynamics of the Zagros-Iran region."

Line 60 "was conducted"

in general ? or for the zagros ? I am lost. Needs a transition.

Reply: - We have rephrased the sentence as follow:

"Numerical modelling of tectonic stress and deformation is generally conducted in two approaches" (Lines 86-87).

Line 66 "it's classified" what does it mean ?

Reply: - The types of stresses

Line 69 "third-order stress" define what are first, second and third order stresses ?

Reply: - The first, second and third order stress is defined in lines 77-82

Line 71-73 "There are numerical studies conducted for tectonic stresses and deformation in Zagro-Iranian region (Austermann & Iaffaldano, 2013; Md & Ryuichi, 2010; Francؚois et al., 2014; Khodaverdian et al., 2015; Vernant & Che´ry, 2006; Yamato et al., 2011)."

wrong. Those studies are thermo-mechanical modelling of collision, topography or subduction, they are not intending to quantify or characterize present--day tectonic stresses.

Reply: - We have modified the sentence (lines 109-131).

Line 76 needs a transition.

Reply: - We have rephrased the sentences for a better flow (lines 109-123).

Line 78 "procided" → "provided"

Reply: - We have corrected the word (Line 130).

Line 100 Add refs to well known papers on the geoydnamic evolution of the Zagros. cites e.G. Agard et al., 2005; Ballato et al., 2011; Mouthereau et al., 2012.

Reply: - We have added the references (Line 155-156).

Line 101 add McQuarrie et al., 2003.

McQuarrie, N., Stock, J.M., Verdel, C., Wernicke, B.P., 2003. Cenozoic evolution of Neotethys and implications for the causes of plate motions. Geophysical Research Letters 30, doi:10.1029-2003GL017992. https://doi.org/10.1029/2003gl017992

Reply: - We have added the reference (Line 157).

Line 102-104 "Zagros muntain belt is also accompanied by a zone of widespread deformation in the form of the high plateaus of Iran. Numerous earthquakes occur in these high terrains due to sustained tectonic activities; hence, these areas are prone to large seismic hazards."

Awkward. The Iran plateau does not belong to the Zagros. The following sentence is a truism. Not clear what is meant to.

Reply: - I have modified and expanded the sentence (lines 158-164).

Line 124 present-day

Reply: - We have added the text (Line 187).

Line 158 "3.2 Crustal Models"

Why using lithosphere thickenss of 100 km ? For calculating GPE you need LAB depths and hypothesis on lithosphere viscosities to compute the tractions. LITHO1.0 is good

LAB model. But there are many seismological others (e.g. Schaeffer and Lebedev, 2013). you can also use other type of geophysical models (e.g. Robert et al., 2015) or the application of Robert's model to derive stresses like Tunini et al. (2017).

Why nowhere in your methodology section you present the LAB depth variations. This is very surprising because it is ciritcal important information to calculate the GPE and tractions in a region where the LAB significantly varies between Arabian shield, Zagros and the Iranian plateau as shown by previous studies.

Tunini, L., Jiménez-Munt, I., Fernandez, M., Vergés, J., Bird, P., 2017. Neotectonic Deformation in Central Eurasia: A Geodynamic Model Approach. J Geophys Res Solid Earth 122, 9461–9484. https://doi.org/10.1002/2017jb014487

Robert, A.M.M., Fernndez, M., Jimnez-Munt, I., Vergs, J., 2017. Lithospheric structure in Central Eurasia derived from elevation, geoid anomaly and thermal analysis. Special Publ 427, 271–293. https://doi.org/10.1144/sp427.10

Reply: - We also tested lithosphere thickness of 150 and 200 km, which did not offer significant changes in style of deformation (Figure S2) (Lines 235-237). In supplementary section, we have shown the comparison of predicted parameters for a LAB depth of 150 km and 200 km for a joint model of CRUST2 and S40RTS (Table S2). We also used Robert et al., 2017 to compute GPE and associated stresses, but again no significant improvement was observed. Hence, we kept the previous crustal and lithospheric models in paper.

Line 167-168 What are the refs for CRUST2.0, CRUST1.0 and LITHO1.0?

Reply: - We have added the references (Lines 239-240).

Line 170 you mean LITHO1.0 ?

Reply: - Yes, I have modified the text (Line 242).

Line 193 "The radial viscosity model, GHW13" Please develop. This is a very critical for your study.

Reply: - We have included an explanation for the radial viscosity models used in our study (lines 266-276).

Line 215 "predicted velocities" Please explain how you obtain the dynamic velocities.

Reply: - We have added explanation for the same in section 3.1 (Lines 222-227).

"We also get the relative plate velocities and strain rates as output from our models. However, to calculate the absolute plate velocities and strain rates, we require absolute viscosity values. We compute the scaling factor for relative viscosities by placing the predicted velocities in a no-net-rotation (NNR) frame, such that $\int (v \times r) dS = 0$ and minimizing the misfit between the predicted dynamic velocities and those from Kreemer et al. (2014). Here v denotes the horizontal surface velocity at position r and S is the area over the Earth's surface (see Ghosh et al. (2013b) for details)."

Line 249 "plate velocities" In theory plate motions are derived from the 3D density and viscosity field. With no convection the velocity field is obtained by assuming some contrasts in the lithosphere thickness and densities. Nothing is said about it. It is well known that together with convection these parameters of viscosities and densities are essential.

Reply: - We have added an explanation for the plate velocities in above comment (Lines 222-227).

Line 279 "the error in predicting plate velocities is higher than in GPE only models." I don't understand the logic here. It is expected that models including both GPE and convection should better fit with the observed plate velocities.

Reply: - Yes, the models including both GPE and mantle tractions better fit the plate velocities. However, in this section, I talked about contributions from mantle convection only, and hence found that the plate velocities predicted from "mantle only" models give higher misfits than "GPE only models". In the next section, I talked about the joint contributions of both, GPE and mantle tractions. I have slightly rephrased the sentence to avoid ambiguity. (Lines 372-373).

Line 281 "we add the" rephrase. You mean you now model the stress field using both contributions ?

Reply: - Yes, I mean to consider contributions from both, GPE and mantle convections models. I have rephrased the sentence (Lines 374-376).

Line 370 "Although we used four tomography models to compute the mantle derived stresses, the stress regimes for all models are found to be similar, with varying magnitudes." This is not suprising considering they are very close and because you not introduce changes in lithosphere thickness and viscosities. Again it would have been interested to chose one tomographic model and impose different LAB models and densities in the lithosphere.

Reply: - I have tested three LAB depths, 100, 150 and 200 km, and have shown the results in supplementary section (Figure S2 and Table S2).

The authors have capitulated the GPE differences with a comparable understanding with existing global tomography models to traceout the stress patterns or deformations in the said study region. This is a good work with possibility of new findings and may be considered for publication after some minor corrections.

1. The authors compared or used Global tomography models. They may consider any regional or local scale geophysical/tomography models for better estimation of the results. I feel there must be more refined sclae results if authors consider this issue.

Reply: - We thank the reviewer for their positive comments. We did try some regional crustal models, as given by Robert et al. , 2017; however it did not offer any significant variations over the crustal models used in our study. Hence, we kept our previous results.

2. Authors have very nicely derived the stress patterns /deformation from numerical analysis. My concern is, though they have picked up the analysis technique, there are also several well renowned methods for understanding the lithospheric or mantle deformation phenomena, one of which is seismic anisotropy or shear wave splitting study. I feel authors must discuss this seismological method. Only a supportive discussion of different methods for deformation analysis will be an added advantage to this paper. I feel no need for analysis and only discussion of these method will be sufficient.

In the most complex regions, like the said study region, Indian counterpart of Himalayn region (and surroundings) is a most complex and deformation region. Authors may discuss and compare the deformation patterns for these regions (repeated collision and subduction tectonics) accordingly. They may refer to certain publications like; Mohanty and Mondal, 2020; Mohanty and Singh, 2021; Mondal and Mohanty, 2021; Mohanty 2023; Singh et., 2016. These references may be cited and discussed only for a better shape of the paper.

Reply: - We thank the reviewer for pointing this out. We have tried to include compression Himalayan-Tibet and Zagro-Iranian region in discussion section and cited some of study above mentioned (lines 554-567).

3. There are certain gramatical errors need to be corrected in the abstract and introduction sections.

Over all, this study has a positive approach towards deformation analysis and may be considered after these above rectifications.

Reply: - We have modified and check the manuscript to avoid the grammatical errors.